# Genomic and Epigenomic Mechanisms of the Interaction between Parasitic and Host Plants

**DOI:** 10.3390/ijms24032647

**Published:** 2023-01-31

**Authors:** Vasily V. Ashapkin, Lyudmila I. Kutueva, Nadezhda I. Aleksandrushkina, Boris F. Vanyushin, Denitsa R. Teofanova, Lyuben I. Zagorchev

**Affiliations:** 1Belozersky Institute of Physico-Chemical Biology, Lomonosov Moscow State University, Moscow 119234, Russia; 2Faculty of Biology, Sofia University “St. Kliment Ohridski”, 8 Dragan Tsankov Blvd., Sofia 1164, Bulgaria

**Keywords:** parasitic plants, host plants, haustorium, resistance, genome, transcriptome, gene expression, mobile RNA, epigenetic variability, siRNA

## Abstract

Parasitic plants extract nutrients from the other plants to finish their life cycle and reproduce. The control of parasitic weeds is notoriously difficult due to their tight physical association and their close biological relationship to their hosts. Parasitic plants differ in their susceptible host ranges, and the host species differ in their susceptibility to parasitic plants. Current data show that adaptations of parasitic plants to various hosts are largely genetically determined. However, multiple cases of rapid adaptation in genetically homogenous parasitic weed populations to new hosts strongly suggest the involvement of epigenetic mechanisms. Recent progress in genome-wide analyses of gene expression and epigenetic features revealed many new molecular details of the parasitic plants’ interactions with their host plants. The experimental data obtained in the last several years show that multiple common features have independently evolved in different lines of the parasitic plants. In this review we discuss the most interesting new details in the interaction between parasitic and host plants.

## 1. Introduction

Approximately 1% of all angiosperms are parasites on other plants and have evolved the parasitic mode of life independently at least 12 times [1,2,3,4]. Parasitic plants are distinguished by their photosynthesis capability (hemiparasites or holoparasites) and by the host plant organ which they penetrate (root feeders or shoot feeders). The most studied groups of parasitic plants are witchweeds (*Striga*), broomrapes (*Orobanche/Phelipanche*), and dodders (*Cuscuta*), probably due to their great economic impact on agriculture [5].

To extract nutrients from the host plants, parasitic weeds use a unique multicellular structure—the haustorium that penetrates into the host organ and forms a connection with its vascular system [6]. Successful parasitism usually causes significant loss of the host plant productivity, decreasing the crop harvest and contaminating it with its own seeds.

The haustorium development occurs in four stages: haustorium initiation (pre-haustorium), host organ penetration, establishment of host–parasite vascular continuity, and the creation of a parasitic sink to draw the host’s nutrients [6]. Pre-haustorium development is initiated upon host detection via some chemical and physical signals. It begins with the establishment of an adhesive structure that glues the parasite to the host organ surface [6,7]. Epidermal cells at this contact site elongate forming haustorial hairs or papillae. The chemical nature of the adhesive glue varies in different parasitic plants [3,6]. Haustorium penetration up-regulates genes encoding the enzymes that loosen the host cell wall [2,3]. Differentiation of the haustorium vascular elements occurs concomitantly with the penetration or soon after when it reaches the host vascular system. The vasculature of the mature haustorium extends nearly perpendicular to the host vascular axis [3]. The haustorium cells metabolize the host nutrients into a parasite-favorable profile, creating a high osmotic potential to draw the nutrients toward the parasite.

Parasitic plants differ in their susceptible host ranges, and the host species differ in their susceptibility to parasitic plants [8]. Furthermore, within the same host species, different varieties could have variable levels of susceptibility or resistance to a specific parasitic weed species. There are few means of crop resistance to parasitic plants. Two main modes of resistance are pre-attachment (before the haustorium attachment) and post-attachment (after the haustorium attachment) to the host plant. An understanding of the molecular mechanism of the host–parasite interaction would be highly helpful in developing new efficient approaches to control the parasitic weeds [1,3]. Though significant progress has been achieved in studying the biology of parasitic plant development, the mechanisms underlying their adaptations to various host plant species are still poorly understood. These adaptations obviously depend on genetic features. However, multiple cases of rapid adaptation in genetically homogenous parasitic weed populations to new hosts strongly suggest the involvement of epigenetic mechanisms. A major task is to understand how the host preferences are determined in different parasitic weeds, including the roles of both genetic and epigenetic features. Recent progress in genomic technologies revealed many new molecular details of the parasitic plants’ interactions with their host plants [9].

## 2. Genomics of Parasitic Plants

### 2.1. Striga

The sequenced genome of *Striga asiatica* was predicted to contain 34,577 genes and retained evidence of at least two whole-genome duplication events as well as some gene losses and gains compared with most photosynthetic plants (Table 1) [10]. By comparing it to the lists of the haustorium gene orthogroups [11] and genes with tissue-specific expression in *Arabidopsis thaliana*, a significant enrichment for tissue-specific orthogroups was found in the *S. asiatica* haustorial genes. In accordance with Yang et al. [11], this pattern was strongest for pollen orthogroups. Thus, evolutionary gain of a haustorium might occur through co-option of genes with tissue-specific gene expression. Relative to common ancestors of *S. asiatica* and its closely related non-parasitic plant *Mimulus guttatus*, ~23% of 10,248 orthogroups showed changes in gene numbers in the *Striga* lineage (647 contractions, 1742 expansions, 456 losses, and 152 gains). The relative age of genes in contracted orthogroups was significantly older than genes in expanded families. Moreover, the expanded gene families showed higher ratios of non-synonymous to synonymous substitutions compared with the contracted gene families. Thus, relatively younger expanded gene families evolved under more relaxed selection pressure, at least at the earlier stages of their evolution, potentially providing a source of genes to encode parasite-specific traits. Probably, a positive selection for some non-synonymous substitutions has occurred at later stages when these genes acquired new function associated with the parasitic lifestyle. The most significant gene family contractions were detected in gene families whose functions were supplemented by the host plants. Like other parasitic plants of the Orobanchaceae family, *Striga* evolved the ability to germinate after sensing strigolactones (SLs), which indicate the presence of a host, by co-opting a divergent subclade of *KAI2* paralogs–*KAI2d* [12]. In *S. asiatica* genome, the *KAI2* gene orthogroup consists of 21 members, including 17 of the *KAI2d* class. Most of them are highly expressed in seeds and seedlings. If different KAI2d proteins have specificity for distinct types of SLs, then the rapid evolution of the *KAI2d* gene subclade likely enabled *Striga* seeds to recognize a wide range of host plants.

In *Striga hermonthica* (the most devastating *Striga* species), twelve clusters of transcripts were defined by distinct expression profiles at different developmental stages: preconditioned seeds; seedlings 48 h after 10 nM strigol treatment; whole seedlings 1 day post infection (dpi) of rice plants; and haustoria attached to rice tissues at 3 and 7 dpi [10]. The early-stage parasitism gene expression was induced by the 2,6-dimethoxy-p-benzoquinone (DMBQ) treatments and by both host (rice) and non-host (*Lotus japonicus*) plant interactions, while the expression of middle- and late-stage genes was not seen in the interaction with non-host plant. Since *S. hermonthica* is able to penetrate into tissues of non-host plants, but not establish xylem connections with *L. japonicus*, these observations mean that early-stage genes are likely to be important for haustorium formation and host penetration, while the middle- and late-stage genes may be required for the xylem connection formation and the host materials acquisition. In situ hybridization analysis confirmed this view. An early-stage peroxidase was exclusively expressed in the intrusive cells at the host–parasite interface, whereas 7-dpi haustoria-specific genes were highly expressed in the hyaline body that functions as a sink for host materials. The middle- and late-stage specific genes included the hydrolase genes during host penetration, transport-related genes during host nutrient acquisition, and signal transduction-related genes during resource allocation. Specifically, enzymes targeting primary cell wall components, such as pectin, were highly up-regulated. Many protease genes were up-regulated at the later stages of infection. It has been hypothesized that parasitic plants employed a pre-existing developmental program to evolve the haustorium [6]. One such program is lateral root formation, as this also creates new xylem connections in roots. All the 18 known lateral root development (*LRD*) genes in *A. thaliana* have orthologs in the *S. asiatica* genome, and 17 of them were detected in the *S. hermonthica* transcriptome. Among these genes, *SLR*, *ARF19*, and *LAX3* were specifically expressed during the early stage of haustorium development. Both *SLR* and *ARF19* are involved in regulating the expression of *LAX3* that encodes an auxin influx carrier, which localizes auxin accumulation during the LRD. Hence, the SLR-ARF19-LAX3 module may be functioning in auxin accumulation during *Striga* haustoria formation. Another putative target of the SLR-ARF19 is the ortholog of *LATERAL ORGAN BOUNDARIES DOMAIN 18 (LBD18)*, which was found to be up-regulated at the early stage of haustorium development. In *A. thaliana*, LBD18 activates cell proliferation in the lateral root primordia. Thus, the *LBD18* ortholog may function in regulating cell proliferation during haustorium formation. Interestingly, orthologs of the two *LRD*-related genes, *ABERRANT LATERAL ROOT FORMATION 4* (*ALF4*) and *ARABIDOPSIS CRINKLY 4* (*ACR4*), were not up-regulated in haustoria, but were up-regulated in the host plants at 1 dpi. It could be suggested that these genes link the interaction between *Striga* and its host. Taken together, these findings suggest that haustorium formation evolved, at least partly, through the recruitment by the parasitic plant of the host LRD programs.

### 2.2. Orobanche/Phelipanche

*Triphysaria versicolor* is a generalist parasite plant that feeds on highly diverse angiosperms, including at least 30 species in 17 families of monocot and eudicots [17]. Transcriptomes from the laser-dissected haustorium samples of *T. versicolor* grown on distantly related host species *Zea mays* and *Medicago truncatula* were sequenced to identify shared and host-specific patterns of gene expression. A large number of unigenes (1124) were shared between the interface transcriptomes of *T. versicolor* grown on both hosts, but absent from the transcriptome of above ground tissues of autotrophically grown *T. versicolor*. These unigenes likely represent the core set of parasite genes that are active irrespective of the host plant species. Many other unigenes were exclusive to a specific host interface (677 for *Z. mays* and 361 for *M. truncatula*), likely representing genes that interact with unique aspects of the host biology. Notably, the unigenes shared in the interaction of *T. versicolor* with both hosts were over-represented for the transcription factor (TF) category and under-represented for the transport category. Therefore, there are TF genes that are specifically active at the parasite–host interface but not expressed in the above ground tissues of autotrophically grown *T. versicolor*. In contrast, most of the transporter gene families that are expressed at the interface are also expressed in the above ground tissues. Among the most highly expressed genes in the shared interface sets were genes for β-expansin and several other cell wall modifying enzymes, a gene encoding a putative AP2-ERF domain TF, and 10 genes with sequence identity to pathogenesis-related proteins in other eudicot species. The putative *β-expansin* gene was highly expressed in *T. versicolor* interface on *Z. mays*, but lowly expressed in *T. versicolor* interface on *M. truncatula*. This apparently host-specific gene expression is notable because of the expansins’ role as cell-wall loosening proteins implicated in the interaction between parasitic plants and their hosts.

In another model species of Orobanchaceae family, *Phtheirospermum japonicum*, haustorium-inducing factors (HIFs) induce the expression of an auxin biosynthesis gene in root epidermal cells at haustorium formation sites, causing cell division and expansion to form a semispherical pre- or early haustorium structure [18]. When exposed to HIFs without a host in close proximity, haustorium formation is induced, but its growth is soon arrested. Thus, the fate of haustorium cells is determined by host availability. Two *P. japonicum* mutants, one dominant and one recessive, cannot regulate cell division and cell fate transitions at the haustorium apex and thus showed defects in host invasion [13]. The mutant lines had elongated haustoria on a medium containing DMBQ. Wild-type (wt) haustoria stopped growth within 2 days after DMBQ treatment, probably to suppress unnecessary growth in the absence of a host. In contrast, mutant haustoria kept elongating even after 3 days of DMBQ treatment, suggesting a defect in such a suppression system. Mutant haustoria often produce a single xylem strand, but it does not connect with root xylem. The putative mutant genes were recently identified for both of these mutants by whole genome sequencing [13]. The draft genome of *P. japonicum* was annotated to contain 30,337 protein encoding genes (Table 1). Two non-synonymous SNPs were identified in the genome sequence of the recessive mutant, one of them located in gene encoding protein phosphatase 2C, while another in the single copy gene encoding a homolog of *A. thaliana* EIN2–key signal transducer of ethylene. The SNP in *PjEIN2* replaces the Arg744 residue with a stop codon, resulting in truncated PjEIN2, likely defective for activating the ethylene signaling in the cell nucleus. In the dominant mutant genome sequence, 36 non-synonymous SNPs were found, of which one occurred in a gene homolog of *A. thaliana ETHYLENE RESPONSE 1* (*ETR1*). This SNP in *PjETR1* causes an Ile to Phe replacement within the first of three hydrophobic regions at the N terminus that function in ethylene binding. In *A. thaliana*, mutations in these hydrophobic regions result in complete loss of ETR1 ability to bind ethylene and in a dominant ethylene-insensitive phenotype. Since both mutants had SNPs in key signaling genes in the ethylene pathway, their mutant phenotypes might be caused by defects in ethylene signaling. Consistent with this assumption, both mutants were ethylene-insensitive for growth inhibition of roots and hypocotyls. Moreover, transformation of a full-length *PjEIN2* sequence restored the normal haustorium phenotype in the recessive mutant. Therefore, the elongated haustorium phenotype was caused by ethylene insensitivity. Elongated haustoria were induced in wt *P. japonicum* by treatments with ethylene signaling inhibitors AgNO_3_ and 1-methylcyclopropene, mimicking the phenotype of *Pjetr1* and *Pjein2* mutants. AgNO_3_ induced elongated haustoria similarly in *S. hermonthica*. Ethylene biosynthesis genes and receptors are also conserved in *S. asiatica* [10]. Therefore, the function of ethylene signaling in haustorium elongation appears to be conserved in the Orobanchaceae. Expression of the auxin-responsive promoter *DR5* coincided with haustorial apical cell proliferation, and more prolonged maxima of auxin response were observed in *Pjein2* haustorium apices compared with wt. Hence, ethylene signaling may be necessary to terminate cell proliferation at the haustorium apex via suppression of the auxin response. In infection assays with *A. thaliana* and rice as host plants, most of the wt haustoria successfully invaded host roots, while invasion levels were significantly lower in *Pjetr1* (by 65–70%) and *Pjein2* (by 91–98%). Non-invaded mutant haustoria failed to develop intrusive cells at the apex and kept elongating around the host surface even after direct contact with the host roots. Thus, ethylene signaling in the parasite appears to be crucial for the haustorium apex cells to differentiate into intrusive cells for host invasion. Apparently, parasitic plants maintain haustorial apex growth until they encounter a position where the host produces ethylene.

When the *P. japonicum* haustoria penetrate the roots of *A. thaliana*, thin cell walls are observed at the site of adherence between the haustoria tips and the host plant tissue, indicating that cell wall digestion occurred at this interface [19]. Similar to the diverse host range of *P. japonicum* as a parasite, it has been noted to have an unusually wide range of compatible partners in interspecies grafting, both as the scion and as the stock plant. It was suggested that mechanisms of these two types of interspecies communication are somehow related. Similar to parasitic association, cross-sections of the graft junctions between *P. japonicum* and *A. thaliana* show xylem continuity and apoplastic dye transport. Therefore, *P. japonicum* is able to achieve tissue adhesion and vasculature connection with members of diverse plant families in both parasitism and grafting. However, the transcriptomes of the *P. japonicum* haustoria during *A. thaliana* root parasitism and those of the *P. japonicum* grafted regions on the *A. thaliana* stems were overall different. Some genes were up-regulated both during parasitism and grafting, including those associated with wound healing processes, such as auxin action, pro-cambial activity, and vascular formation. However, the expression of many other genes was distinct between parasitism and grafting. A member of the *Glycosyl hydrolase 9B* (*GH9B*) gene family, *GH9B3,* is known to be crucial for cell-to–cell adhesion in plant interfamily grafting. Parasitic plants *P. japonicum* and *S. hermonthica* have five and four *GH9B3* genes, respectively, while only two and one such genes are present in non-parasite plants *A. thaliana* and *Lindenbergia philippensis*, respectively. This tendency for the greater number of the *GH9B3* clade genes was also observed in *S. asiatica*. In *L. philippensis*, *GH9B3* was up-regulated at 1 day after grafting both in compatible and incompatible interspecies grafts, but this up-regulation did not continue to increase in subsequent days. By contrast, in *P. japonicum*, the most highly related *GH9B3* homolog was up-regulated at the early stage of both parasitism and grafting and gradually increased in subsequent days. Similar situation was observed in *S. hermonthica*. Some of the other homologous genes of the *GH9B3* clade were also up-regulated in parasitism, but not those of the other *GH9* clades. Therefore, the up-regulation of *GH9B3* homologs at the sites of infection and interfamily grafting seems to be a conserved feature in parasitic plants. Amino acid sequences encoded in *GH9B3* genes show a conserved catalytic domain and O-glycosylation sites. In *P. japonicum, GH9B3* expression was detected at the haustorium cell periphery attaching to the host at 1–2 dpi and in the inside of haustorium at 3–4 dpi. Thus, *GH9B3* appears to function at the parasite–host interface at the early adhesion stage and in xylem formation at later stages. *GH9B3*-knockdown by RNAi did not affect haustorium emergence but significantly reduced the number of successful xylem connections with the host. Therefore, *GH9B3* positively regulates infection processes in *P. japonicum*. Glycosyl hydrolases encoded in the *GH9B3* clade genes are probably β-1,4-glucanases that target glucan chains of cellulose in the primary cell walls. It could well be that *GH9B3*-down-regulated haustoria cannot reach the host vasculature due to insufficient glucanase activity to loosen the host cell wall. In *P. japonicum*, four of the five *GH9B3* genes appear to encode fully functional proteins.

In *P. japonicum* and other parasitic Orobanchaceae, once the haustorium reaches host tissues, the epidermal cells of the parasite haustorium apex differentiate into intrusive cells—the specialized cells for host invasion [20]. In a recent study, differential gene expression was investigated specifically in the intrusive cells at the penetrating tips of the *P. japonicum* haustoria [21]. A total of 3079 differentially expressed genes (DEGs) were detected between the intrusive cells and other parts of the haustorium. Among the DEGs that showed strong and specific expression in the intrusive cells, three were the most notable. A homolog of the *A. thaliana HAESA-LIKE1* (*HSL1*)–*INTRUSIVE CELL-SPECIFIC LEUCINE-RICH REPEAT RECEPTOR-LIKE KINASE1* (*ICSL1*) probably functions as a peptide hormone receptor. *GERMIN-LIKE PROTEIN1* (*GLP1*) encodes a germin-like protein closely related to the *Gossypium hirsutum* ABP19—a superoxide dismutase that regulates redox status. *CONSTITUTIVE DISEASE RESISTANCE1* (*CDR1*) encodes an aspartic protease—a homolog of aspartic proteases that regulate disease resistance signaling in *A. thaliana*. Among the genes that were expressed at higher levels in intrusive cells than other parts of the haustorium, five genes encode subtilisin-like serine proteases (SBTs) involved in biotic interactions. Multiple *SBT* genes were induced in the haustorium at 3 dpi or later, indicating that these *SBTs* are activated after attachment to the host. Transgenic *P. japonicum* plants expressing the coding sequences of proteinase inhibitors Epi10 and AtSPI-1 under the control of intrusive cell-specific *SBT1.1.1* and *SBT1.2.3* promoters showed reduced xylem bridge formation in the haustoria at 5 days after infection of *A. thaliana* roots compared with control plants. Furthermore, expression of Epi10 led to decreased activity of the *ICSL1* promoter in the intrusive cells and diminished auxin responses in the haustoria central region but not around the xylem plate. Collectively, these results suggest that the intrusive cell-specific SBT activities promote the auxin-dependent maturation of haustoria by regulating development of intrusive cells and subsequent xylem bridge formation. In plants, SBTs are known to function in the maturation of plant peptide hormones leading to phenotypic changes such as root elongation, abscission of floral organs, and embryonic cuticle integrity. Apparently, in parasitic plants, some SBTs might evolve for biotic interactions, including parasitism. Since many SBT clades were found to be species-specific, different parasites might recruit SBTs independently to promote parasitism.

### 2.3. Cuscuta

Unlike root parasites, the obligate parasite dodders (*Cuscuta*) establish haustoria on the above-ground parts of their hosts [3,22]. In an early attempt to understand the genetic makeup of dodders, the expressed genes of the two *Cuscuta* species, *C. pentagona* and *C. suaveolens*, were analyzed by high-throughput sequencing [23]. In sequence comparisons between *Cuscuta* and 12 plant species with sequenced genomes, the highest level of similarity was observed with tomato reflecting a close phylogenetic relationship of respective families (Convolvulaceae and Solanaceae) that belong to the same order (Solanales). Notably, some dodder expressed sequence tags (ESTs) showed higher similarities to transcripts of more distant plant species, such as monocots, suggesting that these dodder sequences might have been acquired via horizontal gene transfer (HGT) events. Overall, transcripts of the two dodder species showed high similarity to those of sweet potato, suggesting that few genetic features could account for their distinct lifestyles, likely through the acquisition of new functions by existing genes. Few ESTs that were highly similar between the two dodder species showed significant similarity with transcripts of other parasitic plants but had no significant sequence similarities in non-parasitic species. Obviously, the corresponding genes in the *Cuscuta* genomes could serve as promising targets of RNAi-based approaches to dodder resistance.

The first reference transcriptome of *C. pentagona* was created by de novo assembly of the massive short-read sequencing data on the Illumina HiSeq2000 platform [24]. It should be noted that the *Cuscuta* plants used in that study were indeed members of a closely related species *Cuscuta campestris* [25,26]. The resulting transcriptome encoded 44,758 putative proteins [24]. Gene expression profiles at specific stages of dodder development were not greatly influenced by the host plant species. Gene Ontology (GO) terms “cell wall” and “hydrolase activity” were enriched in the pre-haustoria-specific gene set, while “transporter activity”, “secondary metabolic process”, “response to external stimuli”, “secondary shoot formation”, and “polar auxin transport” were the most enriched terms in the haustoria-specific gene set. Multiple transcripts were up-regulated in haustoria compared with stems and seedlings. The up-regulated DEGs included genes encoding disease, defense, and drug response signals, cell wall-loosening enzymes and modulators, TFs, and auxin related proteins (Table 2). Among the down-regulated DEGs were genes involved in auxin response, such as multiple *INDOLE-3-ACETIC ACID INDUCIBLE*s (*IAA*s), *AUXIN RESPONSE FACTOR*s (*ARF*s), and *SMALL AUXIN UPREGULATED* (*SAUR)-like* genes.

The genes expressed in the interface region between the parasite *Cuscuta japonica* and the host *Glycine max* were studied by analyzing RNA-seq reads obtained from the interface region without physically dissecting either the parasite’s or host’s tissues [27]. Instead of physical dissection, a bioinformatic approach was used to classify the reads into either *C. japonica* or *G. max* transcriptomes. The expression profiles of all DEGs, 3819 and 17,653 of them belonging to *C. japonica* and *G. max,* respectively, were classified relative to the stages of parasitism. Multiple hydrolase enzymes appeared to be involved in molecular interactions occurring in the extracellular space at the parasite–host interface. The expression of cell wall degrading enzyme genes in *C. japonica* occurred at 24–48 h after attachment, when the haustoria penetration proceeded from the host cortex to the host pith, preceding the expression of expansins in *G. max*. The increase in the transcriptional activity of transporter genes was observed at later stage, likely reflecting the increase in sink activity of *C. japonica*. Simultaneous monitoring of gene expression in parasitic and host plants could assist in understanding the coordination of cellular processes between the two plants.

Gene expression in haustoria and stems of two *Cuscuta* species, *C. reflexa* and *C. gronovii*, was analyzed to identify the cell-wall-related genes expressed at the onset and progress of haustorium development [28]. Haustoria were induced either by infestation the compatible host plant *Pelargonium zonale* or in the absence of a host plant by the far-red (FR) light. Unlike the host plant infection that progressed at variable speed, the development of FR light-induced haustoria occurred at a very predictable time pace. Mostly transcripts associated with the cell wall functions were predominant among the haustorium-specific sequences, probably reflecting elevated cell wall remodeling activities in this tissue. The two *XTH* (xyloglucan endotransglucosylase/hydrolase) genes and the *PX-2* (peroxidase) gene showed a more specific expression pattern with ≥80-fold higher expression levels in young haustoria. The xyloglucan-modifying enzymes encoded in *XTH* genes were shown by immunofluorescent microscopy to be located in the cell walls of elongating cells in the areas flanking the haustorial initiation center—the part that is responsible for the swelling of the stem during attachment to a host plant. Furthermore, the xyloglucan endotransglucosylation (XET) activity of XTHs was detected in the parasite throughout haustorium development, mostly at the side facing the host plant. Weaker XET activity was detected in the endophytic part of the haustorium. XTHs are known to function in loosening the plant cell walls by restructuring xyloglucans, allowing turgor-driven expansive cell growth. Seven *XTH* genes in *C. campestris* were found to be highly expressed at the pre-haustorial stage [24], suggesting that the early expression of these genes is a common developmental event in various *Cuscuta* species. Interestingly, the increased expression of a tomato *XTH* gene has been reported to be a possible defense reaction of resistant tomato interaction with *C. reflexa* by tightening the cell walls [29]. It well could be that the infective mechanisms in parasite plants might have evolved through re-purposing of defense pathways that already existed in their non-parasitic ancestors. The results obtained indicate that changes to cell walls are essential in the formation of the haustorium. Therefore, cell wall genes and *XTH* genes in particular might prove effective gene silencing targets for the control of *Cuscuta* in agriculture.

**Table 2 ijms-24-02647-t002:** Common pathways and underlying genes in various parasitic plants revealed by whole-genome gene expression analyses.

Parasite	Common Parasitism Pathways	Common Parasitism Genes	References
*Striga asiatica*	SL recognition, cell wall modification, auxin signaling, lateral root development	peroxidase, hydrolase, protease, auxin related, lateral organ boundaries domain (*LBD)*, lateral root development (*LRD)*	[10]
*Striga hermonthica*	proteolysis, auxin signaling, oxidation–reduction processes, protein phosphorylation, transport, cell wall modification, serine-type peptidase	*SLR*, *ARF*s, *LAX3, LBD18,* oxidase, aspartyl protease, serine carboxypeptidase, peroxidase, LRR N-terminal domains, kinase, receptor protein, transporter, pectate lyase, pectin methylesterase inhibitor, cellulase	[10,11]
*Triphysaria versicolor*	cell wall modification, proteolysis, oxidation–reduction processes, protein phosphorylation, transport, serine-type peptidase	β-expansin, AP2-ERF, PR proteins, pectate lyase, aspartyl protease, peroxidase, LRR N-terminal domains, serine carboxypeptidase, serine-type peptidase, subtilase, kinase, receptor protein, transporter, pectin methylesterase inhibitor, cellulose	[11,17]
*Phelipanche aegyptiaca*	proteolysis, oxidation–reduction processes, protein phosphorylation, transport, cell wall modification, serine-type peptidase	aspartyl protease, LRR N-terminal domains, serine carboxypeptidase, peroxidase, kinase, receptor protein, transporter, pectate lyase, pectin methylesterase inhibitor, cellulase	[11]
*Phtheirospermum japonicum*	ethylene and auxin signaling	*EIN2*, *ETR, GH9B3,* LRR receptor-like kinase, oxidase, protease	[13,19,21]
*Cuscuta campestris* *	carbohydrate metabolism, cell wall, solute transport, phytohormones, protein degradation, RNA biosynthesis, protein modification, polyamine metabolism, cell cycle, chromatin, cytoskeleton	pectin lyase, pectin methylesterase, xyloglucan endotransglucosylase, polygalacturonase, cellulase, *ARF*s, expansins; LRR kinases, receptor-like kinases, *SAUR*, peptidase, SL biosynthetic enzymes, nutrient transporters, *KNAT*s, *WRKY*, *YABBY*, *AUX*	[14,24,30,31]
*Cuscuta reflexa*	cell wall remodeling	pectin acetylesterase, glycoside hydrolase, xyloglucan endotransglucosylase/hydrolase, peroxidase	[28]
*Cuscuta australis*	terpenoid biosynthetic process, nitrate assimilation, regulation of signal transduction, response to auxin, DNA methylation	pectin esterase, receptor-like kinases, transport proteins, subtilisin-like proteases, ABC transporter, α/ß-hydrolase (SL receptor?)	[15]
*Santalum album*	cell wall, mitochondria, ribosome, protein turnover, auxin, cytokinin, GA, ABA, JA, ethylene, brassinosteroids	pectinesterase, polygalacturonase, pectin lyase, xyloglucan endotransglucosylase/hydrolase, expansin, β-D-xylosidase, pectin lyase, and glycosyl hydrolase	[32]
*Thesium chinense*	cell wall, oxidation, reduction, proteolysis, terpene synthesis, fatty acid metabolism, flavonoid synthesis, carbohydrate and sugar transport, auxin signaling, very long chain fatty acid biosynthesis	pectin methylesterase, phenylalanine ammonia-lyase, *SHORT INTERNODES/STYLISH* (*SHI/STY*), *PIN-LIKES7* (*PILS7*), *SUPPRESSOR OF G2 ALLELE SKP1* (*SGT1*), *LATERAL ORGAN BOUNDARIES DOMAINs* (*LBDs*), *ACYL-COA OXIDASE 2* (*ACX2*), *ACYL-COA SYNTHETASE5* (*ACOS5*), *ACOS7, KNAT6, SCARECROW-LIKE* (*SCL*), *WUSCHEL-RELATED HOMEOBOX* (*WOX*), *PILS6*, *ARF9*	[33]
*Taxillus chinensis*	metabolism and environmental adaptation, amino sugar and nucleotide sugar metabolism, mineral absorption, defense response, proteolysis	MYB TFs, WRKY TFs, bHLH TFs, ribosomal proteins, ubiquitin, E3 ubiquitin-protein ligase, disease resistance proteins	[34]

* Mistakenly identified as *Cuscuta pentagona* in [24].

The first high-quality reference genomes for *Cuscuta* species were assembled relatively recently for *C. campestris* [14] and *C. australis* [15] (Table 1). The genome sequence of *C. campestris* contains 44,303 predicted gene loci that were supported by transcriptome data and/or sequence similarity to known sequences [14]. Of the 1440 genes typically conserved in plants, 82.1% were present in the *C. campestris* assembly, and 59% of them in a duplicated form. Of the assembled sequences, 46.2% were identified as transposon-derived, mostly LTR retrotransposons. The high proportion of gene duplicates with a low synonymous substitution rate (d*S*) value and the recent LTR-retrotransposon proliferation pointed to a recent genome duplication event. More than thousand gene families conserved in all other tested dicots were found to lack orthologs in *C. campestris.* A total of 1736 genes were lost in *C. campestris* relative to *Ipomoea nil*—the closest available non-parasitic relative of *Cuscuta*. Most of the genes that distinguish *C. campestris* from non-parasitic plants appeared to be highly conserved across land plants, supporting the assumption that their function has become obsolete due to the parasitic lifestyle. In the search for putative HGT events, 64 novel candidates from at least 32 different donor sequences were found in the *C. campestris* genome, with a functional bias towards defense reactions or unknown functions in their hosts. Most of them could be traced back to the preferred host orders Fabales and Caryophyllales suggestive of HGT between host and parasite. Some of these transferred genes showed strong up- or down-regulation in haustorial vs. control tissues, suggesting that their gene products play a role in the infection process.

More than half of the ~273 Mbp genome sequence in *C. australis* appeared to consist of repetitive elements [15]. Similar to *C. campestris*, LTR retrotransposons are the dominant type of repeats in *C. australis*. Of the total 13,981 gene families that were identified in *C. australis* and seven reference plant species, 1256 had been significantly contracted and 478 significantly expanded in *C. australis* compared with phylogenetically related autotrophic plant species. Of about 12,000 conserved gene orthogroups identified in the seven reference species, 1402 were absent in *C. australis* (Table 1). In *Solanum lycopersicum* and *I. nil*, genes of these orthogroups are preferentially expressed in leaves and roots, consistent with the absence of these organs in *Cuscuta*. Overall, massive loss of genes was found in gene pathways controlling leaf and root development, nutrients uptake from soil, photosynthesis, flowering time, and defense against pests and pathogens. It was found that 1124 genes underwent positive selection after the divergence between *Cuscuta* and their closest autotrophic relative *I. nil*, including those associated with hormone responses, DNA methylation, regulation of transcription, and cell wall-related metabolism. Among these genes, 115 were found to be preferentially expressed in pre-haustoria/haustoria, including genes encoding a pectin esterase, receptor-like kinases, TFs, a serine carboxypeptidase, and transport proteins (Table 2). Five positively selected genes from the expanded gene families were found to be preferentially expressed in haustoria, including a gene that encodes a putative α/β-hydrolase highly similar to *Nicotiana sylvestris* DAD2/DWARF14 which is an SL receptor in autotrophic plants. Thus, neo-functionalization of α/β-hydrolase genes might be involved in the evolution of parasitism both in the stem parasites *Cuscuta* and in the root parasites *Striga* and *Orobanche* [12].

An in vitro system for inducing haustorium development outside an intact host was developed and used to examine the host-dependence of the haustoria formation in *C. campestris* [30]. When lateral shoot segments of *C. campestris* were pressed with a stack of glass slides and subjected to blue light irradiation for 72 h, two types of haustoria were induced. The haustoria with protruding search hyphae were regarded true haustoria, while the conical-shaped haustoria without searching hyphae were termed pseudohaustoria. Under these conditions, when *C. campestris* shoot segments did not attach to the host, elongation of axial cells and search hyphae was observed in the true haustoria, but the search hyphae did not differentiate into xylem hyphae. These true haustoria were similar to those observed just after penetration into the host stems. No visible alterations in development were observed in the true haustoria when the dodder shoot segments were exposed to the phytohormone mixtures known to induce xylem vessel differentiation in other angiosperms. Hence, the inability to induce differentiation in the absence of host tissue was not due to a lack of phytohormones derived from the host plants, but some other host-derived factors were needed to induce the xylem vessel differentiation in *Cuscuta* haustoria. When the lateral shoot segments of *C. campestris* were overlaid with fresh rosette leaves of *A. thaliana*, and then pressed with a stack of glass slides, haustoria invaded the host tissue and the final process in xylem vessel differentiation was observed at the contact area with the host xylem. This in vitro xylem differentiation was quite similar to that observed during *C. campestris* parasitization on the intact host plants. RNA-seq was used to examine the transcriptional regulation of haustorium development during the time-course of haustoria penetration into the host tissue. RNA-seq libraries were prepared from tissue samples taken at 57 h after induction (hai), when most haustoria had penetrated the host rosette leaf, and 87 hai, when differentiation of search hyphae into xylem hyphae had occurred. True haustoria that did not contact the host tissue were also used at the same time points, and epidermal and cortical cells of *C. campestris* at 0 hai served as a “no-haustorium” control. A total of 15,277 DEGs were identified in the haustorium compared with the “no-haustorium” sample. Of these DEGs, 4239 were shared between the four haustorium conditions. Consistent with previous gene expression studies [24,28], genes encoding functionally annotated proteins for carbohydrate metabolism, cell wall, and solute transport, as well as phytohormones, protein degradation, and RNA biosynthesis, were up-regulated in the haustorium. At 57 hai, genes encoding proteins for cell wall metabolism, phytohormones, protein modification, and secondary metabolism were significantly enriched among the DEGs up-regulated in haustoria that penetrated the host rosette leaf. At 87 hai, genes related to phytohormones, polyamine metabolism, and RNA biosynthesis were up-regulated in these haustoria. Thus, gene expression is dynamically regulated during haustoria penetration into host tissue and during their further development. When the expression of orthologous genes known to be involved in the development and proliferation of vascular stem cells, was compared at 57 hai between haustoria that penetrated the host tissue and true haustoria that were induced in the absence of the host tissue, most of these genes were up-regulated in both samples, indicating that haustoria acquired the potential for differentiation into xylem cells even in the absence of host tissue. At 87 hai, genes encoding proteins involved in the promotion of xylem vessel cell formation were not up-regulated in the haustoria that did not penetrate the host tissue. On the contrary, these genes (*VND7*, *MYB46*, *MYB83*, *CESA4/IRX5*, and *CESA7/IRX3*) were up-regulated in the haustoria that penetrated the host leaves in the in vitro system and in the haustoria produced by *C. campestris* shoots 54 h after coiling around an intact stem of *A. thaliana*, when search hyphae contacted the host xylem. These data show that contact of search hypha with the host xylem induces the up-regulation of a *VND7* ortholog and its downstream target genes in *C. campestris* resulting in the formation of xylem vessel cells in the haustorium.

Though initiation and progression of haustorium in *Cuscuta* depend on signals from the host plant, a host-free haustorium induction by a combination of far-red (FR) light and tactile stimuli [28] or blue light [30] can be used to study the haustorium development in a more uniform and predictable experimental paradigm. A FR-induction was employed in a recent study to correlate gene expression patterns with characteristic morphological traits during successive stages of haustorium development [31]. In this experimental setup, apical portions of the *C. campestris* stems developed haustoria that bear many of the morphological and molecular characteristics of naturally developing haustoria. After 1–2 days, a slight bump appeared on the surface. It showed on the cross-sections that epidermal cells underwent a strong elongation perpendicular to the surface and haustorium initials appeared in the region between the cortex and the vascular tissue. This early “swelling stage” (SWE), soon became more pronounced and formed a readily visible structure of ~1–2 mm that stuck to the surface it faced—the “attaching stage” (ATT). In cross-sections at this stage, the newly formed pre-haustorium with its meristem and elongation zones was visible. The epidermal cells facing the surface were palisade-formed and dark purple stained owing to their high pectin content. Then, the haustorium emerged in the center of this lateral structure with search hyphae at its tip that sometimes protrude from the surface. In the host-free system, this was the final stage that can be observed—the “penetrating stage” (PEN), while on a host plant, the action of degrading enzymes causes a rupture in the infected tissue and the intrusion of the haustorium. Stem sections below and above the region where haustoria developed (non-infective stems or “niS”) were used as reference sites. The most profound changes in gene expression occurred during the transition between niS and SWE, with 3440 and 1500 DEGs up-regulated and down-regulated, respectively. By contrast, only about half as many DEGs were differentially regulated in the subsequent stages: 2326 in ATT relative to niS and 2554 in PEN relative to niS. In both cases, the majority of DEGs were up-regulated (1934 and 2145, respectively). Of these DEGs, 1012 were common to the three haustorial stages. The highest number of stage-specific DEGs was observed at SWE (2783), followed by PEN (943), while only 192 DEGs were specific for the transitory stage ATT. Hence, largest changes in gene expression occur in the very beginning of haustorium development. More DEGs in SWE than the two later stages support the involvement of numerous regulatory processes in the earliest step of haustoriogenesis. Among these DEGs were genes of expansins that contribute to cell wall loosening and cell expansion. Auxin-related biosynthetic enzymes and transporters, as well as 11 of the 22 alpha-class expansins were up-regulated in SWE. Other DEGs that were enriched in SWE, including those related to cell cycle, cytoskeleton and cell wall, likely coordinate meristem development and organogenesis. Beyond this initial stage, more subtle changes in gene expression may indicate that once the haustorial development has been started, relatively small adjustments are needed to advance the process further.

### 2.4. Santalaceae

*Santalum album* (sandalwood) is one of the economically important plant species in the Santalaceae family due to its use in the production of highly valued perfume oils. Sandalwood is also a root hemi-parasite that can produce photosynthetic products, but needs to obtain some of its water and simple nutrients from other plants through haustoria. In nature, at least 300 species, including *S. album* itself, can act as hosts of sandalwood tree. To identify genes and main pathways involved in haustorium development in *S. album*, the transcription profiles of its haustoria were studied at the pre-attachment (1 to 10 days after haustorium initiation) and post-attachment (10 to 20 days after haustorium initiation) stages [32]. Twenty-day-old non-haustorial seedling roots were collected as the control (R) sample. The majority of the *S. album* transcripts were expressed in all three tissue samples, indicating that very similar sets of genes underlie the haustorial and root development. Only 90, 41, and 224 transcripts were preferentially expressed in roots, pre-attachment haustoria, and post-attachment haustoria, respectively. Many genes associated with cell wall metabolism and mitochondrial respiratory chain function, were highly expressed in pre-attachment haustoria relative to roots and in post-attachment relative to pre-attachment haustoria. Most genes encoding cell wall remodeling proteins, such as pectinesterase (PE), polygalacturonase (PG) precursor, pectin lyase-like superfamily protein (PL), xyloglucan endotransglucosylase/hydrolase (XTH), and expansin-like (EXL), were up-regulated in haustoria relative to roots. Several genes encoding proteins in this functional category, such as β-D-xylosidase (XYL), pectin lyase (PL), and glycosyl hydrolase superfamily protein (GH), were further up-regulated in post-attachment relative to pre-attachment haustoria. Thus, cell wall remodeling is probably involved in the growth and differentiation of haustoria. Moreover, DEGs associated with the mitochondrial respiratory chain function were strongly up-regulated in developing haustoria, suggesting that high activity of mitochondrial respiratory function is necessary to provide energy for the rapid growth of haustoria and establishing host–parasite connections. Genes encoding ribosomal proteins were also significantly up-regulated in pre-attachment haustoria relative to roots and still more up-regulated in post-attachment haustoria. A similar up-regulation in haustoria was observed for DEGs encoding enzymes of protein turnover. These results show that ribosome biogenesis and protein turnover increase when the haustorium develops and invades the host root. The most abundant class of TFs that were differentially expressed between the haustoria and roots was GRAS. Of the GRAS TFs, 49 (96%) were up-regulated in pre-attachment haustoria relative to roots and 11 (30%) were further up-regulated in post-attachment relative to pre-attachment haustoria. Some of them were specific to haustoria, suggesting that GRAS TFs may be critical for haustorium development. A total of 191 DEGs in pairwise comparisons were associated with plant hormone biosynthesis, metabolism, and signal transduction pathways, including auxin, cytokinin (CK), gibberellin (GA), ABA, jasmonic acid (JA), ethylene (ET), and brassinosteroid (BR). The largest group (91 DEGs) was involved in auxin transport and signal transduction. The majority of auxin-related genes showed higher expression in pre-attachment haustoria relative to roots, but many of them were down-regulated in post-attachment compared with pre-attachment haustoria. Three genes of auxin influx carriers (*LAX*) and four genes of auxin efflux carriers (*AEC*) were up-regulated in pre-attachment haustoria relative to roots. All 14 DEGs encoding proteins of polar auxin transport (*BIG*s) were up-regulated in pre-attachment haustoria, but then down-regulated in the post-attachment haustoria, suggesting that high auxin level might be needed for the haustoria initiation.

Like *S. album*, *Thesium chinense* is a facultative root hemi-parasite belonging to the family Santalaceae. It is commonly distributed in the grasslands of Eastern Asia and, similar to *S. album*, can parasitize a broad range of plant species. In a recent work, the detailed transcriptomic and metabolomic changes were studied in the samples of *T. chinense* haustoria collected from their native habitat [33]. Haustoria initiated from fully mature roots of *T. chinense* invade the host root tissues with their tips. Differential expression analysis revealed 2265 DEGs between haustoria and roots of *T. chinense*. Of these DEGs, 801 were up-regulated and 1464 down-regulated in haustoria (Table 2). Genes encoding pectin methylesterase and phenylalanine ammonia-lyase showed up-regulation in haustoria. Three auxin-related genes, *SHORT INTERNODES/STYLISH* (*SHI/STY*), *PIN-LIKES7* (*PILS7*), and *SUPPRESSOR OF G2 ALLELE SKP1* (*SGT1*), and two *LBD* genes were among the top 10 most significantly up-regulated genes in haustoria. Thus, auxin response could be one of the shared mechanisms in haustorial formation among parasitic plants. The genes related to very long chain fatty acid (VLCFA) biosynthesis, *ACYL-COA OXIDASE 2* (*ACX2*), *ACYL-COA SYNTHETASE5* (*ACOS5*), and *ACOS7* were also up-regulated in haustoria. Besides being components of the seed storage fats, VLCFAs play a key role in regulation of cell proliferation and differentiation in plants and are involved in polar auxin transport to determine the cell polarity in lateral root development. By co-expression gene network analysis, *ACOS7*—the VLCFA biosynthesis gene expressed in haustoria—was shown to be a highly interconnected hub gene in a module containing several DEGs between haustoria and roots, such as the lateral root developmental genes, *LBD25* and *KNAT6*, and another VLCFA biosynthesis gene, *ACX2*. Other genes related to lateral root meristem formation, *SCARECROW-LIKE* (*SCL*) and *WUSCHEL-RELATED HOMEOBOX* (*WOX*), and genes related to auxin signaling, *PILS6* and *AUXIN RESPONSE FACTOR 9* (*ARF9*), were also in this module. Collectively, these data show that strongly associated VLCFA biosynthesis genes and auxin-responsive lateral root developmental pathway genes act as a key gene network in the developmental reprogramming of *T. chinense* haustorial formation. Indeed, pentacosanoic acid was found to be highly abundant in haustoria probably due to the up-regulation of the VLCFA biosynthesis genes.

### 2.5. Loranthaceae

*Taxillus chinensis* (loranthus) is a facultative stem hemiparasite plant of Loranthaceae family that attacks other plants in multiple families. Recently transcriptome profiles of haustoria development in *T. chinensis* were analyzed [34]. Of the total 14,295, 15,921, and 16,402 genes that were expressed in fresh seeds, early haustoria, and adult haustoria, respectively, 12,888 genes appeared to be common, 3749 DEGs were detected between early haustoria and fresh seeds, and 4139 DEGs—between adult haustoria and fresh seeds (Table 2). Among these DEGs, 1543 up-regulated and 1086 down-regulated were common to early and adult haustoria. Of the 863 TF genes that were expressed in the haustoria, 174 showed differential expression. Most of these TF DEGs were common between early and adult haustoria. Compared with seeds, 9 ethylene-responsive (ER), 5 MYB, 6 WRKY, and 11 bHLH TF genes were up-regulated in both early and late haustoria. Some TF genes were specifically up-regulated in early haustoria, including 1 ER, 4 WRKY, and 5 bHLH TF genes, while 10 ER, 5 WRKY, 6 bHLH, and 4 MYB TF genes were up-regulated in adult haustoria only. Of the total 1194 ubiquitin genes, 81 were differentially expressed in haustoria relative to fresh seeds. Among them, 17 DEGs were down-regulated in early haustoria, and 11 were also down-regulated in adult haustoria. 38 ubiquitin genes were up-regulated in early haustoria, and 26 of them were also up-regulated in adult haustoria. Notably, six ubiquitin genes were gradually up-regulated along with the haustoria development, including three polyubiquitin, two ubiquitin-40S RP, and one E3 ubiquitin-protein ligase genes. Of the 226 genes encoding disease resistance proteins (DRPs) in haustoria, 94 were differentially expressed. Of these 94 DRP DEGs, 87 were up-regulated in haustoria, of which 51 were common to early and adult haustoria; seven DRP DEGs were up-regulated and eight DRP DEGs down-regulated in adult relative to early haustoria.

### 2.6. Scrophulariaceae

*Monochasma savatieri* is a perennial root hemiparasite herb, which is an ingredient in traditional Chinese medicine for curing urinary and upper respiratory tract infections. In a recent study, RNA profiles in *M. savatieri* roots were studied before and after successful parasitism on the host plants *Gardenia jasminoides* [35]. The transciptomes of *M. savatieri* roots were compared at the two key time points—8 weeks after sowing (WAS) when the parasitic relationship with the host was still not established and 16 WAS—after establishing parasitic relationship with the host. The growth and development of *M. savatieri* was significantly promoted after parasitizing the host. Of the all four pairwise comparisons between these plants and the respective control *M. savatieri* plants that were grown in the absence of the host plants, the largest number of DEGs (46,424) were observed in 8 WAS vs 16 WAS *M. savatieri* parasitizing plants comparison. In parasitizing plants at 16 WAS, 27,098 DEGs were up-regulated and 19,326 DEGs down-regulated compared with 8 WAS, when the parasitic relationship with the host plant was still not established. Furthermore, in this comparison, DEGs encoding 66 TFs were identified. These TFs may have regulatory roles in the establishment of parasite–host relationship. Among these TFs, the MYB family contained the largest number of DEGs, seven putative MYBs being up-regulated upon establishment the parasite–host relationship. Interestingly, eight putative TFs of the WRKY family were down-regulated, suggesting the role of negative regulators of the establishment of parasite–host associations. In the plant hormone signal transduction pathway, the highest number of DEGs was involved in auxin signal transduction, and their expression levels were increased after establishing the parasite–host relationship. Thus, similar to other parasitic plants, the auxin pathway may play an important role in the parasite–host association.

### 2.7. Rafflesiaceae

Species in the flowering plant clade Rafflesiaceae represent the most extreme form of parasitism achieved by plants [36]. Gigantic flowers of these species emerge directly from their hosts and exhibit no obvious plant body. In its vegetative stage, the parasite resides inside the host as a thread-like filament known as the endophyte. The endophytic strand is only a single cell layer wide (uniseriate) and cells typically divide perpendicularly to its axis. It lacks any discernible cell differentiation and cytologically resembles the undifferentiated embryo. During transition to flowering, the uniseriate endophyte enters a multiseriate stage and then forms a tear-shaped mass of parasite cells lacking organ differentiation, known as the protocorm. As the protocorm grows, it begins a process of cellular differentiation. Some of the cells at the periphery differentiate into xylem that establishes the connectivity between the host and the parasite. Cells internal to the host–parasite interface at the protocorm front flatten and at later stages pull away from the rest of the protocorm to create a cushion for the floral bud as it pushes through the hard woody tissues of the host vines. The flowers mimic the putrid smell to attract the flies that pollinate them. Genetic studies in Rafflesiaceae suggested an unusual genome evolution, including the complete loss of their plastid genome and multiple HGT events. These features make Rafflesiaceae of particular interest for further comparative genomic investigation. The first genome assembly for an endophytic plant parasite has been done recently [16] (Table 1). *Sapria himalayana* Griff. is a species of Rafflesiaceae that parasitizes three distantly related *Tetrastigma* species (Vitaceae) in Southeast Asia. The genome of this species appeared to have an unusually low GC content (~24%), mostly due to the high proportion (~89.6%) of the AT-rich repeat elements, while the smaller compartment consisted of the gene-rich scaffolds with higher GC content (mean 41.2%). The high number of predicted genes (~55,000) appeared to be mostly represented by a small number of abundant orthogroups consisting of transposable elements (TEs). Thus, 736 (10.9%) orthogroups that contain TE-like domains account for 35,136 (82.6%) of the validated *Sapria* genes. Additional gene expansion was identified in 710 (10.5%) medium-sized orthogroups (<100 gene copies each). Genes in these orthogroups are involved in various functions, including chromosome organization, DNA metabolism, and the cell cycle. Consistent with its extraordinary reduction in morphology and the life cycle, an unprecedented gene loss was found to occur during the *Sapria* genome evolution. Nearly half (44.4%) of the 10,880 orthogroups that are universally conserved across eudicots, are absent from the *Sapria* genome–more than in any other group of angiosperm parasites. Of the conserved genes lost in *Sapria*, 13.2% (n = 642) are also lost in *Striga* and *Cuscuta* parasitic clades. Within these convergently reduced functional categories, a far greater number of genes were lost in *Sapria* than in other parasites, especially of photosynthesis-related genes. The most extreme example of this tendency in *Sapria* is the complete loss of the plastid genome and nearly complete loss of nuclear genes that regulate plastid organization and function. Higher losses were also found in functional categories not previously seen in other parasitic plants, such as biosynthesis of ABA, protein degradation, and purine metabolism. Thus, 18 of 27 genes associated with ABA biosynthesis in *A. thaliana* were lost in *Sapria*. In the protein degradation pathway, significant gene losses occurred in the ubiquitin-proteasome-mediated protein degradation and the endopeptidase Clp-mediated protein lysis. These losses may be caused by the reduced requirement for nutrient recycling and abiotic stress response. In the purine metabolic pathway, only one homolog of nucleoside diphosphate kinase (*NDPK1*) and two homologs of nucleoside-triphosphatase (*AYP1* and *AYP2*) are retained in *Sapria* versus four and six homologs, respectively, in *A. thaliana*. These reductions may reflect reduced requirements for natively produced metabolites due to their uptake from the host plant. The majority of genes in *Sapria* are very compact, showing fewer introns than *Genlisea aurea*—a carnivorous plant species that has the smallest angiosperm genome known [37]. In *Sapria*, at least 18.7% of the genes have lost all introns that are present in both of its closest free-living relatives, *Manihot* and *Populus*. Its highly compact genes (intron length <150 bp) are significantly enriched for housekeeping functions, such as DNA and RNA metabolism, stem cell maintenance, and reproduction. This contrasts sharply with the free-living plants whose intron size is largely independent of gene function. This feature of housekeeping genes may convey a selective advantage of more efficient transcription for parasites that rely on their host for energy and chemical resources. On the other hand, a substantial proportion of the genes in *Sapria* contain unusually long introns. For introns longer than 1 kb, 74% of the total length consist of TEs. A wide phylogenomic analysis of 55 plant species, including three transcriptomes from Rafflesiaceae, eighteen transcriptomes from their obligate hosts Vitaceae, and 33 published genomes spanning the angiosperm phylogeny, identified HGT events in 568 *Sapria* genes and pseudogenes from 81 orthogroups corresponding to 1.2% of uniquely aligned genome sequences. These HGTs range from 100 bp to 16.5 kb in length, and 62% of them are intergenic. Introns were detected in all but two HGT genes where the donor sequences contained introns, supporting previous findings that the uptake of naked foreign DNA is the primary source of HGT.

RNA sequencing analyses at the whole genome scale were widely used in the last years to identify genes potentially involved in the parasite–host plant interaction. Some of these genes appeared to be expressed at the parasitic stage in haustoria in various parasitic species, probably reflecting convergent features in their evolution (Table 2).

## 3. Genetic Basis of Susceptibility and Resistance in the Host–Parasite Plant Interaction Revealed by Genomic Studies

### 3.1. Rice–Striga hermonthica

In many cases, resistance to parasitic weeds is multigenic and largely environment-dependent. It often involves several mechanisms that are relatively weak and tend to break down in the new environment or in the presence of new variety of the same parasite. For many years, the use of resistant cultivars was limited due to poor understanding of the genetic basis of susceptibility and resistance. An early whole genome study compared gene expression profiles between the two rice cultivars that showed different resistance to the witchweed *S. hermonthica* [38]. In the resistant cultivar Nipponbare, resistance to *S. hermonthica* was characterized by the development of hypersensitive response (HR) around the site of parasite attachment that led to necrosis and the inability of the parasite to penetrate into the host xylem. Another cultivar used, IAC 165, showed high susceptibility to *S. hermonthica* infestation. Changes in gene expression at 2, 4, and 11 dpi were studied in both cultivars by hybridization to the rice Affymetrix oligonucleotide array. In the susceptible interaction with IAC 165, the parasite radical attached to the host root at 2 dpi, while at 4 dpi the haustorium penetrated the host root cortex and endodermis to form parasite–host xylem connection. The parasite then developed rapidly so that by 11 dpi it had 2–4 leaf pairs and fully differentiated haustoria. In a sharp contrast, most parasites that attached to the Nipponbare roots failed to develop. The initial stage of infection looked similar to that on IAC 165; by 2 dpi parasites attached to the host root and began to penetrate the root cortex. However, by 4 dpi, rings of necrosis were visible around parasite attachment sites. By 11 dpi, these rings of necrotic tissue enlarged, and the parasites died. In the resistant interaction between Nipponbare and *S. hermonthica*, 1653 rice genes were either up- or down-regulated by more than twofold, while in the susceptible interaction between IAC 165 and *S. hermonthica,* 2079 genes showed changed expression. Of these DEGs, about 200 were unique in each of the two modes of interaction. In the resistant interaction, greater numbers of genes were up-regulated within the functional categories “gene expression/protein fate” and “plant defense responses” than in sensitive interaction. Conversely, in the sensitive interaction, more genes were up-regulated within the categories “metabolism” and “cellular transport”. Many genes and pathways among those up-regulated in the Nipponbare resistance response to *S. hermonthica* were common with defense responses against fungal and bacterial pathogens, such as genes encoding endochitinases (PR-3), glucanases (PR-2), and thaumatin-like proteins (PR-5).

### 3.2. Cowpea–Striga gesnerioides

In cowpea *Vigna unguiculata*, resistance to the witchwood *Striga gesnerioides* exhibits two types of defense: (i) HR at the site of parasite attachment, followed by the death of the parasite within 3 to 4 days; (ii) growth arrest of the parasite at the tubercle stage [39]. The type of resistance appears to be dependent both on the host genotype and the parasite race. A gene named *RSG3-301* that conferred resistance to *S. gesnerioides* race SG3 was isolated and sequenced. Its sequence predicted an R protein with a coiled-coil protein-protein interaction domain, a nucleotide-binding site, and a leucine-rich repeat domain (CC-NBS-LRR). Silencing of *RSG3-301* expression in the resistant cowpea cultivar B301 leads to susceptibility to race SG3 but does not affect resistance to other races of the parasite, underscoring the specificity of the resistance response. It was suggested that race-specific resistance to *S. gesnerioides* in cowpea is an example of effector-triggered immunity (ETI) in which intracellular proteins are activated by the parasite effectors [40].

Cowpea cultivar B301 was originally identified as resistant to all races of *S. gesnerioides* known at the time; however, a new hyper virulent race SG4z overcame that resistance [41]. The differential response of B301 to parasite races was used to examine genome wide changes in gene expression in the same genetic background between susceptible (B301-SG4z) and resistant (B301-SG3) interactions. To this end, RNA samples from two time points were analyzed by hybridization to a custom-made cowpea microarray. The time points chosen corresponded, in resistant interaction, to the earliest evidence of a HR (6 dpi) and to the complete host cortical root browning and necrosis of the parasite (13 dpi). In susceptible interaction, the host–parasite vascular connection was established by 13 dpi, and successful parasite growth was evident. In the resistant interaction at 6 dpi, the expression of 111 genes was significantly altered in cowpea roots compared to control (uninfected) roots. At 13 dpi, the number of significantly altered genes increased to 2102, including 52 genes whose expression was already altered at 6 dpi. At both time points, more than half of the DEGs were up-regulated. The large increase in the number of DEGs at 13 dpi could be explained by the propagation of the HR and subsequent biosynthetic and physiological changes. In susceptible interaction, 1944 genes changed expression at 13 dpi, more than half of them (1089) down-regulated. Of the DEGs found, 923 genes were common between resistant and susceptible interactions at 13 dpi, while 1179 and 1021 DEGs were unique in the resistant and susceptible interactions, respectively. In the resistant interaction at 6 dpi, genes encoding TFs and signal transduction proteins, defense-related proteins, and proteins involved in cell wall biogenesis were among the most up-regulated. At 13 dpi of resistant interaction, genes encoding TFs, signal transduction proteins, proteins of the cellular energy metabolism and those involved in developmental regulation were among the most up-regulated. In susceptible interaction at 13 dpi, genes encoding defense-related proteins, auxin regulator proteins, and proteins involved in cell wall biogenesis were among the most down-regulated. Thus, in the resistant interaction, the visible HR is accompanied by the up-regulation of genes involved in signal transduction and biosynthetic processes associated with formation barriers to prevent parasite penetration such as cell wall biogenesis and lignification, as well as processes leading to programmed cell death at the host–parasite interface. Some of these genes and pathways are repressed in the susceptible interaction, suggesting that the parasite is targeting specific components of the host defense. To identify the effector that suppress resistance in the B301 cultivar, transcriptome profiles were compared between SG4 and SG4z haustoria [42]. One transcript showed no detectable expression in SG4 but the high expression in SG4z. The encoded 195 amino acid protein has a 25 amino acid predicted extracellular (apoplastic) targeting signal peptide at the N-terminus and four LRRs, one of them located adjacent to the signal sequence and three arranged in tandem near the C-terminus. The predicted 3D structure of the protein indicated that three tandem LRRs form a potential protein interaction domain. The respective gene has been sequenced and designated as *Suppressor of Host Resistance 4z* (*SHR4z*). The gene is present in other races of *S. gesnerioides* but not expressed in them. The SHR4z role as a secreted effector involved in manipulating host root innate immunity was confirmed in experiments with its ectopic expression in the host plant roots. The host target of SHR4z appeared to be a BTB-BACK domain-containing protein VuPOB1 with a high level of similarity to *A. thaliana* AtPOB1 and *Nicotiana benthamiana* NbPOB1 proteins. Both have been previously identified as essential ubiquitin-protein ligases (E3s) involved in plant immune responses and are thought to ubiquitinate PUB17, a downstream component in the pathway, for subsequent degradation. *VuPOB1* transcript levels in B301 roots parasitized by SG4z and SG4 showed a substantial but transient increase at 3 dpi compared with 0 and 10 dpi. When the VuPOB1 expression was silenced in B301 roots via RNAi, SG4 infection was accompanied by significantly fewer HR events and more tubercle swellings events compared with control roots. Thus, VuPOB1 appears to be a necessary and significant component of innate immunity. It was further confirmed when the effects of ectopically overexpressing VuPOB1 on *Striga*–host interaction was examined in the normally successful parasitism of B301 by SG4z. Unlike the normal successful interaction, B301 overexpressing VuPOB1 showed more HR events and less tubercle swellings. Even more significantly, no cotyledon expansion events were observed. These data clearly showed that VuPOB1 functions as a positive regulator of HR and host immunity in cowpea.

The identification of SHR4z, whose unusually active expression in SG4z allows the suppression of the HR triggered by parasite infection, illustrates the evolution of new components in the arms race between host plants and their parasite plants. In this particular case, host-driven selection of variants capable of overcoming host immunity within the parental SG4 population was likely facilitated by the repeated cropping of the multi-race resistant cultivar B301 over several decades. The occurrence of a mutation that increased expression of *SHR4z* allowed the mutant parasite individuals to complete their life cycle. Because of autogamous reproduction in *S.gesnerioides*, this mutation then was rapidly fixed within the respective SG4 subpopulation giving rise to the SG4z sub-race. The use of secreted effectors to modulate host resistance is a widespread phenomenon in plants interaction with a wide variety of pathogenic organisms, such as microbes, fungi and nematodes. The work of Su et al. [42] shows that this mechanism extends to host plant–parasitic plant interactions.

### 3.3. Sunflower–Orobanche cumana

Similar to a known role of a membrane receptor protein, CuRe1 in the resistance response to *Cuscuta* infestation in tomato [43], a receptor protein (HAOR7) was found to mediate resistance of sunflower to the race F of *Orobanche cumana* [44]. Like *CuRe1*, *HaOr7* encodes an LRR-LRP kinase that is truncated in susceptible sunflower cultivars. The fully functional HAOR7 protein originates from a wild *Helianthus* accession and confers resistance during the early stages of interaction with *O. cumana*. HAOR7 protein from the resistant sunflower line is a 1005 amino acid protein that contains an extracellular LRR domain in the N-terminal region, a transmembrane domain, and an intracellular kinase domain in the C-terminal region. Thus, *HaOr7* encodes a receptor-like membrane protein that leads to signal transduction. In susceptible sunflower lines, alleles of this gene contain a stop codon that results in a truncated protein of 591 amino acids that lacks the intracellular kinase domain, thus preventing signal transduction. The resistant allele of *HaOr7* was shown to completely prevent connection of *O. cumana* to the resistant line due to incompatible attachment.

The sunflower hybrid cultivar EMEK3 has high, long-term resistance to *O. cumana*. In a recent study, its resistance mechanism was studied by transcriptome sequencing [45]. Development of the *O. cumana* seedlings was arrested after attaching and attempting to invade the roots of the resistant cultivar. The disruption of the parasite penetration into the host roots and the subsequent deterioration of the parasite seedlings were accompanied by a darkening of host and parasite tissues at the penetration point. Histological examination showed that the intruding *O. cumana* cells were blocked at the cortex of the resistant cultivar roots and could not reach the endodermis. As evidenced by safranin staining, lignification of cell walls occurred in resistant cultivar, preventing connection of the parasite to the host vascular system. Comparative transcriptome analysis of *O. cumana*-infested vs -non-infested sunflower roots of the resistant cultivar EMEK3 (E), five resistant accessions (R bulk), and five susceptible accessions (S bulk) were used to identify DEGs associated with sunflower resistance. Three DEGs appeared to be common in various resistant sunflower interactions with *O. cumana* and were identified as genes encoding β-glucanase, β-1,3-endoglucanase, and ethylene-responsive TF 4 (ERF4). In the EMEK3 roots post-infestation with *O. cumana*, genes encoding β-1,3-endoglucanase and β-glucanase were up-regulated about 2.5-fold*,* while the gene encoding ERF4 was down-regulated about three-fold relative to EMEK3 roots post-infestation without *O. cumana*. These findings indicate activation of the plant’s innate immune system in which the recognition of PAMPs induces a HR and the accumulation of pathogenesis-related (PR) proteins, such as β-glucanases. It has been shown that β-glucanases, which are able to degrade the cell wall β-glucan, are involved in resistance to *Orobanche crenata* in pea [46] and in sunflower resistance to *O. cumana* [47]. Down-regulation of the *ERF4* gene post-infestation could be connected with the known role of ERF family TFs in regulating defense responses [48]. In *A. thaliana*, *erf4* mutant has been shown to be resistant to *Fusarium oxysporum*, while the transgenic plants overexpressing *ERF4* were susceptible. The down-regulation of the *ERF4* gene in roots of the resistant sunflower could mean that, similar to *A. thaliana*, in the sunflower, response to biotic stress is negatively regulated by ERF4. This view was further supported by a recent study that identified *ERF* as a candidate gene for *O. cumana* resistance in sunflowers [49]. In the resistant sunflower cultivar EMEK3, two lines of defense against *O. cumana* infestation may act successively. After *O. cumana* attachment to the sunflower roots, a typical pathogen-triggered immunity (PTI) response is induced that down-regulates *ERF4* thus counteracting its suppressive activity towards PR genes, including those encoding β-glucanases. The up-regulated β-glucanases then break down the parasite cell walls, releasing effectors that induce the second line of the plant defense–effector-triggered immunity (ETI). A physical barrier is then created by the accumulation of lignin and other phenolic compounds around the penetration area, and the parasite seedlings fail to establish a connection with the host vascular system, and eventually die of necrosis.

### 3.4. Host Plants–Cuscuta

Few plants have defense mechanisms that can successfully prevent *Cuscuta* infestation. Most studies until recently have explored the causes of haustorial inhibition based on macro-tissue structure or physiological changes, without clear molecular targets and mechanisms. In a recent study, a comparative analysis of transcriptomes and metabolomes was performed between *C. japonica* plants interacting with a host plant *Ficus microcarpa* and with a non-host plant *Mangifera indica* [50]. *C. japonica* exhibited distinct morphological features after attachment to *F. microcarpa* and *M. indica*. The dodder seedlings could wrap around *M. indica*, but the haustoria did not fully penetrate the bark of *M. indica*, the stem color of the dodder became black, and the dodder eventually died after 15 days. In contrast, haustorial growth was stimulated by the interaction between the dodder stem and the susceptible host, as indicated by retention of the green coloration of the dodder stem. These results suggest that, unlike *F. microcarpa*, *M. indica* can prevent the dodder absorbance of nutrients through haustoria. A total of 348 DEGs at 24 h after attachment (haa) and 1123 DEGs at 72 haa were identified in pairwise comparisons between susceptible and resistant interactions. Interestingly, at 24 haa, most of the up-regulated genes were detected in dodders on the non-host plants, while at 72 haa, most of the up-regulated genes belonged to the dodders on the host plants. A KEGG pathway enrichment analysis revealed that “flavonoid biosynthesis” was significantly enriched by DEGs at 24 haa, while “plant–pathogen interaction”, “arginine and proline metabolism”, and “MARK signaling” pathways were enriched by DEGs at 72 haa. In all, 13 KEGG pathways were specific for genes in the susceptible interaction, including “valine, leucine, and isoleucine biosynthesis” and “phenylalanine metabolism”, while 12 unique KEGG pathways in resistant interaction included “arginine biosynthesis” and “carbon fixation in photosynthetic organisms”. Collectively, these results show that many metabolites and signal pathways are differently expressed between dodders attached to susceptible and resistant hosts, probably being one of the factors underlying the host selection in dodders.

Tomatoes are generally susceptible to *Cuscuta* infestations. However, the cultivated tomato (*Solanum lycopersicum*) is resistant to *C. reflexa* and exhibits a HR to attempted penetration by *C. reflexa* haustoria [43]. A heat-resistant but protease-sensitive factor that induced ROS production and ethylene synthesis in *S. lycopersicum,* but not in susceptible plants, has been detected in extracts of *C. reflexa* and five other *Cuscuta* species. This active factor appeared to be common to *Cuscuta* species but absent from plants outside this genus. Multi-step purification and mass spectrometry (MS) analysis failed to identify this active factor but the data obtained suggested that it is associated with a small, potentially modified (O-glycosylated?) peptide. An LRR receptor-like protein (LRR-RLP) was putatively identified to mediate specific effects of the *Cuscuta* factor and has been termed CuRe1 (*Cuscuta* receptor 1). In *S. lycopersicon* genome, *CuRe1* encodes a typical LRR-RLP with an N-terminal signal peptide for export via the endoplasmic reticulum, a large LRR ectodomain with 30 to 32 LRRs and 18 potential N-glycosylation sites, a single transmembrane helix, and a short cytoplasmic tail. Only truncated forms of *CuRe1* are present in *Solanum pennellii*. The biological function of CuRe1 was validated by stably transforming *CuRe1* constructs into *S. pennellii* and *N. benthamiana*, both susceptible to *C. reflexa* infestation. Transformed lines gained responsiveness to *Cuscuta* factor and showed increased resistance to *C. reflexa*. The exact nature of the *Cuscuta*-specific ligand of CuRe1 remained unknown for some time. It has been noted in a later study that high amounts of this factor were released from the cell wall fractions of *Cuscuta* by treatments with pectinases [51]. The purified active fractions were partially sequences by tandem MS and blasted against the translated transcriptome database of *C. reflexa* [28]. A perfect hit glycine-rich protein (CrGRP) consisted of 116 amino acids with an N-terminal targeting sequence of extracellular localization. A cysteine-rich 21 amino acid fragment of CrGRP, termed as crip21, retained full activity of the full-length CrGRP. In accordance with the presence of CuRe1-specific ligand activities in different species of *Cuscuta* [43], close homologs of *CrGRP* were found in *C. campestris* and *C. australis*. Both CcGRP and CaGRP contain peptide motifs with a high sequence similarity to the *C. reflexa* Crip21, and both peptides CcCrip21 and CaCrip21 showed full activity in the CuRe1-dependent ethylene induction tests. In general, GRPs appeared to be widely distributed all over the plant kingdom. In cultivated tomato (*S. lycopersicum*), a homolog with an amino acid similarity of 57% to CrGRP was found, but the respective peptide motif SlCrip21 showed only residual activity in the CuRe1-dependent ethylene induction test. Thus, in tomato and other plants, *GRP* genes may play other roles not related to CuRe1-dependent defense responses. The biological role of the full-length protein GRP in dodders is also unclear. Crip21 peptides from *I. nil*, *Nelumbo nucifera*, and *Lactuca sativa* were active in the CuRe1-dependent ethylene production test. Therefore, the CuRe1-receptor system might evolve in tomato as a perception system for molecular patterns of non-self, such as invading dodder parasites. *Cuscuta* GRPs comprise the molecular patterns with characteristic Cys-residues in the Crip21 that mark them as parasites to host plants with the cell surface receptor CuRe1. However, full resistance against *Cuscuta* requires more than CuRe1 immunity in tomato. In search of the additional layers of defense against *Cuscuta*, cell wall lignification genes were studied in a Heinz tomato cultivar resistant to the dodder infestation [52]. When different tomato cultivars were infested with *C. campestris*, lignin accumulation in the stem cortex was observed in the resistant, but not in the susceptible cultivars. The local lignification in the stem cortex appeared to prevent haustorium penetration, and the dodder attachment on the resistant cultivars. An RNA-seq analysis at different time-points after *C. campestris* attachment on a susceptible tomato cultivar H1706 showed that maximal transcriptional changes occurred at 4 days after attachment (daa), probably involving genes of the early response to *C. campestris* infestation. In a comparative analysis of infested susceptible and non-susceptible cultivars, 113 DEGs were selected as the key regulatory gene candidates determining the resistance level in different tomato cultivars. Consistent with the observed lignification in resistant cultivars upon *C. campestris* infestation, known lignin biosynthesis genes were among these DEGs. To narrow down the list of potential candidates and identify upstream master regulators of the lignin-based resistance, the search was further focused on TFs and membrane or cytosolic receptors that might sense signals from *C. campestris*. Three such genes selected were two TF genes related to *AP2* and *SlMYB55*, and a gene encoding a CC-NBS-LRR protein. All these three DEGs were significantly down-regulated upon infestation in susceptible cultivars as well as in the pairwise comparison of susceptible vs resistant cultivars. However, their expression remained essentially unchanged upon the attempted infestation in resistant cultivars. Therefore, active expression of these genes upon infestation might play role in increased resistance to *C. campestris*. When the coding regions of *AP2-like* and *SlMYB55* were transiently expressed in the susceptible tomato cultivar, induction of stem lignification was observed without *C. campestris* infestation, indicating the possible roles of these TFs in regulating some critical enzymes in the lignin biosynthesis pathway. Consistent with this suggestion, the AP2-like protein was renamed into LIF1 (lignin-inducing factor 1). In contrast, plants expressing the *CC-NBS-LRR* gene showed no lignin accumulation phenotype. However, when the responses to *C. campestris* infestation were compared between susceptible tomato plants before and after *CC-NBS-LRR* expression, only plants expressing *CC-NBS-LRR* showed induced lignification upon *C. campestris* attachment. Thus, *CC-NBS-LRR* was renamed into *CuRLR1* (*Cuscuta R-gene for Lignin-based Resistance 1*). When the expression of *AP2-like*, *SlMYB55*, and *CuRLR1* in plants of resistant tomato cultivar was down-regulated by the virus-induced gene silencing, these resistant plants became more susceptible to *Cuscuta* infestation. On the opposite, when *SlMYB55*, *LIF1*, and *CuRLR1* were first overexpressed in the susceptible tomato cultivar plants and then *C. campestris* strands were attached to these plants, lignin accumulation in the cortex was induced that blocked haustorium penetration and made plants more resistant to *C. campestris*. Probably, CuRLR1 serves as a cytoplasmic receptor that recognized some unknown *Cuscuta* signals and further increases lignin deposition in the resistant plants. These unknown signals appear to be large heat-sensitive proteins (30–100 kDa) distinct from the previously identified small *Cuscuta* signal 11 kDa glycine-rich protein recognized by CuRe1 [51]. Two defense responses to *Cuscuta* infestation in resistant tomato plants may be parts of a multi-layer defense mechanism, similar to defense systems against microbial pathogens where MAMP-triggered immunity (CuRe1-mediated response) and effector-triggered immunity (CuRLR1-mediated response) act as two perception steps in the plant defense system. SlMYB55 and LIF1 probably act in parallel and serve as positive regulators of lignin biosynthesis.

In a recent study, gene expression changes were studied in the interface between *C. campestris* and tomato as the host plant [53]. The three time points of haustorium development were chosen as follows: early—the haustorium has just contacted the host; intermediate—the haustorium has developed searching hyphae, but has not formed vascular connections; and mature—the haustorium has established continuous vascular tissue between host and parasite. To identify the genes involved in haustorial development, the protruding regions of haustoria were collected and used for RNA extraction and transcriptome analysis. To study the host responses to *C. campestris* parasitism, few layers of tomato cells that surround the penetrating haustoria were also collected. Overall, the gene expression patterns at the early and intermediate stages were distinct from those at the mature stage. The gene set highly expressed at the early haustorial stage was highly enriched for genes involved in responses to hormones, stress and the far-red light. These genes are likely involved in the haustorium initiation process, since both physical contact with the host and far-red light environment are important for haustorium induction in *Cuscuta* species. Many central hub genes in this gene set encode protein kinases or enzymes involved in cell signaling. Among these central hub genes was a gene encoding homeobox-leucine zipper protein similar to the TF homeobox 7 (HB7) of *A. thaliana*–a known negative regulator of ABA response. The gene set highly up-regulated at the mature stage was enriched for genes involved in root radial pattern formation, in accordance with author’s recent discovery that *C. campestris* utilizes the root developmental program during haustorium organogenesis [54]. Many central hub genes in this gene set are involved in cell wall modification, including expansins and several pectin methyl-esterase inhibitors (PMEIs). Notably, the key regulator of *C. campestris* haustorium development, transcription factor *LATERAL ORGAN BOUNDARIES DOMAIN* 25 (*CcLBD25*) previously identified [54] was highly expressed both at the early and mature haustorial stages. Similar expression profiles were observed for many genes encoding enzymes that catalyze degradation or modification of pectin, such as pectin lyase (PL), pectin methyl esterase (PME), pectin methyltransferase (PMT), and pectin acetyl esterase (PAE). Also, genes encoding an auxin efflux carrier and an auxin-responsive protein were highly expressed both at the early and mature haustorial stages. Furthermore, among the central hub genes in this gene set was an ethylene responsive element binding factor 1 (ERF1), which is the member of the ERF/AP2 domain-containing family of transcription factors. To further validate the function of the *C. campestris* genes *CcHB7*, *CcPMEI*, and *CcERF1*, transgenic tomatoes with hairpin RNAi constructs targeting these genes were constructed. Many haustoria growing on *CcHB7* RNAi, *CcPMEI* RNAi, and *CcERF1* RNAi transgenic tomatoes stopped their penetration in the cortex region. Furthermore, the host cortex cells surrounding the haustoria were enlarged and had a very loose cell wall structure. These haustoria were not able to form vascular connections with their hosts and easily detached from the host stems. Notably, since these plants were in a host-induced gene silencing system, some successful haustorial connections were necessary for the siRNAs to be transferred from the host to the parasite. Indeed, a successfully connected haustorium was often observed followed by several abnormal haustorium attachments. The overall phenotypes of dodder plants grown on transgenic plants showed very few haustorial connections compared to those growing on wt tomato plants. Thus, down-regulation of the *C. campestris* genes, *CcHB7*, *CcPMEI*, and *CcERF1* interferes with haustorium development and diminishes *C. campestris* parasitism. On the other side of the host–parasite interface, the gene expression patterns in the host cortex tissues surrounding *C. campestris* haustoria were distinct from those in the control tomato cells. The tomato genes that were highly expressed at the early haustorial stage involved those associated with defense responses, including *SlWRKY16*—a negative regulator gene of the lignin-based resistance response. Genes of several LRR receptor-like protein kinases, including the homolog of *CuRe1* [43] were also highly expressed at this stage. Many genes involved in ethylene signaling, known to play a role in plant defenses against various biotic stresses [55] were also in this gene set. Thus, the host genes that are highly up-regulated at the early stage are involved in perceiving parasite signals. *Pathogenesis-Related protein 1* (*PR1*) was one of the top central hub genes in the tomato gene set expressed at the early stages of haustorium development. Unlike the host genes that were highly expressed at the early stage only, another gene set consisted of the host genes that were up-regulated at the early stage and gradually decreased their expression throughout parasitism. Many genes in this set encode LRR receptor-like kinases or NBS-LRR proteins (NLRs). Among the genes in this set, four genes encode proteins involved in ethylene signaling, including an ethylene-responsive TF, an ethylene-responsive proteinase inhibitor, and an ethylene-inducing xylanase receptor. Thus, ethylene may play a role in plant responses against parasitic plants. *NLR*s are common disease resistance genes (*R* genes) known to be involved in biotic stress detection, including various plant pathogens and pests [55,56]. Therefore, these NBS-LRR proteins are potentially involved in the process of detecting parasitic plant signals. To test the hypothetical roles of the tomato genes *PR1*, *CuRe1-like*, and *NLR* in responses against the *Cuscuta* parasitism, respective knockout mutants were produced using CRISPR-Cas9 technology. The *CuRe1-like* null mutant plants appeared to be very vulnerable to pathogens and insect herbivores, and produced very few seeds in the greenhouse. Thus, *SlCuRe1-like* might play a role in the various kinds of plant defense responses. In the absence of *C. campestris*, *PR1* and *NLR* mutant plants did not differ in their overall phenotypes from wt tomato plants of the parental line. However, the differences were quite evident when these mutant plants were infested by *C. campestris*. On the wt tomato plants, the searching hyphae entered the host cortex and linked to the host xylem and phloem, but no gross changes of the overall host stem structure occurred. In contrast, mutant tomato plants appeared to be more sensitive to the parasite attack and showed hypertrophy symptoms, such as abnormal plant outgrowth caused by cell enlargement at the haustorium attachment sites. The vascular connections between host and parasite, especially the xylem bridges, were enlarged. *C. campestris* haustoria not only penetrated and formed vascular connections with *NLR* knockout plant vascular tissue, but also changed the overall host stem vascular tissue arrangement, causing a reduction in the secondary xylem in the region of haustorium penetration. This phenotype indicates an increased vulnerability to *Cuscuta* infection.

Collectively, these data clarify, to a significant extent, the molecular mechanisms that underlie variable susceptibility of different host plants to the infestation by the same parasite plant (Figure 1).

## 4. Epigenomics of the Parasite–Host Plant Interaction

### 4.1. DNA Methylation

Among the remarkable adaptations of the obligatory root parasites broomrapes is the sensitivity of the seeds to molecules produced by the host roots that are required for the parasite seeds to start germination [1]. In *Phelipanche ramosa*, seeds require a minimum of 4 days of conditioning at 21°C to allow optimal germination in response to the germination stimulants [57]. Even being conditioned, broomrape seeds are still unable to germinate in the absence of the host germination stimulants produced by surrounding host roots. While the role of SL as a germination stimulant has been known for many years, almost nothing was known about the early molecular events governing SL-induced seed germination until the major role of *PrCYP707A1*, an ABA catabolic gene, in *P. ramosa* seed germination response to the synthetic SL analogue GR24 has been established [57]. GR24 treatment of conditioned seeds triggered a rapid up-regulation of *PrCYP707A1* during the first 18 h, followed by an 8-fold decrease in ABA levels after 3 days. The concomitant application of ABA or a specific inhibitor of CYP707A prevented germination of conditioned seeds. Thus, germination occurs after a dormancy release of the conditioned seeds mediated by the GR24-dependent up-regulation of the *PrCYP707A1*. It was noted that the *PrCYP707A1* could not be up-regulated by GR24 during a minimal 4-day period following imbibition. Apparently, some repressive mechanism prevents the *PrCYP707A1* to be activated during this period, which is switched off later. Indeed, it was demonstrated that cytosine demethylation during the conditioning period is a key step in seed germination by controlling the *PrCYP707A1* response to SL [57]. When the *P. ramosa* seeds were conditioned in a standard medium, the minimum period required for the seeds to germinate in response to GR24 was, as expected, 4 days. In these conditions, the global DNA methylation level remained stable during the first 2 days, and then significantly decreased from day 3 onwards to reach nearly half of the initial level on day 4, concomitant with the seed gained ability to germinate in response to GR24. Interestingly, when seeds were conditioned in ABA-containing medium for 7 days, the DNA methylation level also decreased to reach a value representing about half of the initial level, though the induced seed germination was inhibited. When the seeds were conditioned in the presence of the demethylating agent, 5-AzaC, the minimum conditioning period required to reach the optimal germination rate upon GR24 treatment was reduced to 3 days. DNA methylation level in seeds treated with 5-AzaC decreased continuously from day 1 just after imbibition finally reaching the low value similar to the level observed in standard medium after 7 days. When seeds were conditioned in the presence of 5-AzaC, the induction the *PrCYP707A1* expression by GR24 became possible one day earlier, i.e., after 3 days of conditioning. Thus, sensitivity to GR24 and the *PrCYP707A1* inducibility seemed to depend on the global DNA methylation level. Two CpG islands were found in the *PrCYP707A1* promoter region, at −2183/−1708 bp and −706/−406 bp relative to the transcription start site. The methylation status of the first CpG island (at −706/−406 bp) has not changed between day 0 and day 7 of the conditioning period, while the second CpG island showed a significant demethylation in a 78-bp region between −1839 and −1762 bp. Interestingly, mostly CHH sites were affected by this demethylation. Collectively, these results suggest that the CHH-specific methylation of a short DNA sequence within the promoter CpG island prevents the *PrCYP707A1* gene induction, while its demethylation after 4 d of the conditioning period switches of this repressive mechanism.

### 4.2. Chromosome Organisation

The genus *Cuscuta* consists of three subgenera, *Monogynella*, *Grammica*, and *Cuscuta*, and includes about 200 species of parasitic plants and is exceptional among angiosperms with respect to variation in genome size and chromosome organization [58]. Holoploid genome size (1C) varies from 0.27 Gbp in *C. australis* [15] to 32.12 Gbp in *C. indecora* [59]. Thus, more than 100-fold variation in genome size is observed in *Cuscuta* species—an extraordinary range for a single genus. The existence of very small and very large genomes in *Cuscuta* indicates that the parasitic lifestyle did not play a major role in genome size evolution. The main driving force behind the genome size increases in *Cuscuta* was the amplification of repetitive DNA. In *C. campestris*, 108 genic and 42 non-genic DNA regions appeared to be acquired via HGT events [14,60]. Moreover, species with both monocentric (subgenera *Monogynella and Grammica*) and holocentric (subgenus *Cuscuta*) chromosomes are present within *Cuscuta*. The chromosome structure and the centromere-specific epigenetic histone modification marks have been recently investigated in twelve *Cuscuta* species representing all three subgenera and different clades within the most diverse subgenus *Grammica* [58]. Holocentric chromosomes were confirmed for all three species in the subgenus *Cuscuta*—*C. europaea*, *C. epithymum,* and *C. epilinum,* as evidenced by the absence of primary constrictions, typical chromosome arrangement during mitotic anaphase, and chromosome-wide distribution of histone H3 phosphorylated at serine 10 (H3S10ph). The transition to holocentricity in the subgenus *Cuscuta* was associated with substantial changes in the centromeric chromatin. It was demonstrated that in *C. europaea*, CENH3, which is a centromere protein in most organisms, either has lost its function or acted in parallel to an additional CENH3-free mechanism for kinetochore positioning. Strikingly, the phosphorylation of threonine 120 of histone H2A (H2AT120ph) known to be a universal centromere marker in plants, was not detected in the holocentric *Cuscuta* species. Probably, this histone modification was lost in the subgenus *Cuscuta*. By contrast, chromosomes in species of the other two subgenera possessed primary constrictions and showed the expected single domain localization of both H3S10ph and H2AT120ph, as well as the centromere-specific histone H3 variant CENH3.

### 4.3. Small Non-Coding RNAs

In the *Monochasma savatieri* hemi-parasite plant, 198 differentially expressed miRNA genes (DEmiRs) were detected in pair-wise comparisons between the control free-living plants and those parasitizing on the host plants of *Gardenia jasminoides* at 8 weeks after sowing (WAS), when the parasitic relationship with the host was still not established, or at 16 weeks after sowing (WAS), after establishing parasitic relationship with the host [35]. In comparison with control plants, the numbers of down-regulated and up-regulated miRs in the parasitizing plants at 16 WAS were 29 and 49, respectively. Comparison of parasitizing plants at 8 and 16 WAS showed 55 miRs to be down-regulated and 107 miRs up-regulated at 16 WAS. Most DEmiRs showed larger expression differences between the control and parasitizing plants at 16 WAS than at 8 WAS. The correlation analysis between DEGs and DEmiRs demonstrated that most co-differentially expressed miRNA-target pairs were observed in the comparisons between plants with established parasitic relationships, implying that most dynamically expressed miRNAs and their targets play significant roles in the post-parasitization stage. Furthermore, the target genes enriched for “cell junction”, “extracellular region”, “membrane-enclosed lumen”, and “symplast” functions were differentially expressed in *M. savatieri* plants before and after establishing parasitic relationships with the host plants. These data show that both symplastic and apoplastic transport of nutrients from the host plant occurs in *M. savatieri*.

The sRNA profiles were studied in the *C. campestris* tissues pre- and post-exposure to a host plant but without direct contact to the host [61]. These tissues included germinating seedlings, apical regions from shoots that were in search of a new host (20–80 cm distant to closest infection site to minimize the proportion of host-derived miRNAs), and early and late stages of artificially induced host-independent haustoria. Similar to the other Solanales members, a majority of miRNAs in *C. campestris* were of 21nt in size, followed by miRNAs of 20nt. In *O. aegyptiaca* also the 21-nt miRNAs were predominant, but unlike in *C. campestris*, the 22-nt miRNAs accounted for a larger proportion than the 20-nt miRNAs. Overall, the sRNA analyses of both root and shoot parasites point toward an importance of 24-nt sRNAs that, at least in *C. campestris*, are TE derived. The sRNA populations, however, appear to display dynamic changes during haustoriogenesis. A core group of 24 miRNAs was shared between *C. campestris* and other Solanales members. Thirty-five miRNA families were common between *C. campestris* and seven or more photosynthetic Solanales species, including highly conserved plant miRNA genes like *miR156*, *miR172*, and *miR390*. Interestingly, *C. campestris* appeared to lack several miRNAs that are involved in the development and resistance at different levels. Notably, miR482, a miRNA family that targets *NBS–LRR* genes to produce phased siRNAs and that is highly conserved in Solanales, was not found in *C. campestris*. The same applied to the following miRNA families: miR394, a contributor to the regulation of leaf morphology through the targeting of a *F-box* gene, miR395 that regulates sulfate concentration by targeting the ATP sulfurylase family in *A. thaliana*, miR162 and miR403 which target the RNA silencing members DCL1 and AGO2/3, respectively, and miR477 and miR827 that are involved in the control of immunity by targeting the phenylalanine ammonia-lyase and ubiquitin E3 ligase genes, respectively. Other typical large and conserved miRNA families like miR169 or miR399, which are involved in regulation of stress-induced flowering, had a reduced number of members in *C. campestris*. Thus, *C. campestris* might retain miRNA families important for the regulation of developmental processes and lose miRNA families that control the defense responses and organ development. From the 17 miRNA families that showed differential expression between haustorium and non-infective tissue, eight were exclusively present or up-regulated more than two-fold in haustoria. Only two miRNA families, miR845 and miR156/7, showed up-regulation at the later haustorial stage compared with the earlier one. Highly conserved miRNAs accounted for 43.85% of all identified target sites in genes. Of the total targeted events, 70.87% corresponded to TEs, suggesting a crucial role of miRNAs in controlling TE expression in *Cuscuta* haustoria. Seventeen miRNA families targeted exclusively TEs, while 16 targeted exclusively genes, and 16 appeared to target both TEs and genes. The conserved miRNA targets included a majority of TF superfamilies, two homologs of *AGO1*, and homologs of *ARF6* and *ARF8*. Some cleavage sites were unique to *C. campestris*, suggesting neo-functionalization events. These included a remorin protein homolog that was identified as a target of the conserved miR164 (which targets CUP-SHAPED COTYLEDON TFs in *A. thaliana*), and a plasma membrane intrinsic protein targeted by miR845 (known to target TEs in *A. thaliana*). Four novel miRNAs targeted several TEs, and one of them, miR21, had also a genic target—the RAB-GTPase GDP-dissociation inhibitor. Overall, the analysis of the miRNA targets in *C. campestris* showed that, while most conserved miRNAs still exert their expected functions, some of them underwent neo-functionalization. New miRNAs that were identified are presumably associated with the regulation of parasitic relationship with host plants. The parasitic lifestyle of *C. campestris* had considerable consequences on the configuration of its genome. Multiples genes known to be conserved in other Solanales were lost in *Cuscuta* genomes [14,15]. Among them are genes involved in nutrient uptake, development, and defense. Likewise, a number of miRNAs that are important for the overall development and nutrient sensing in normal green plants appeared to be absent in *Cuscuta*. Some of these miRNAs might be lost because their targets were either lost or did not require regulation due to different lifestyle of the parasite. Other miRNAs might be repressed in the parasite due to its different modes of nutrient supply. Illustrative examples are miR395, known to be involved in the control of sulfate assimilation in normal green plants, and miR482, which in Solanaceae targets several *NBS–LRR* defense genes. Several miRNAs regulating different TF families and some genes associated with transporters and heat shock proteins showed a more enhanced expression in haustorial tissues. In *Cuscuta,* these genes are differentially expressed between haustoria and seedlings. It could be suggested that specific expression of these miRNAs in haustoria adds additional layer of tissue-specific transcriptional regulation.

### 4.4. Exchange of Informational Molecules between the Host and Parasitic Plants

The informational molecule exchange between the host and parasitic plants has been first observed by Roney et al. [62]. When the dodder *C. pentagona* has been grown on the pumpkin *Cucurbita maxima* as the host plant, transcripts of three pumpkin genes, *CmNACP*, *CmSUTP1*, and *CmWRKYP*, were detected in dodder tissues by RT-PCR and sequencing of the RT-PCR products. When the RNA samples of the dodder grown on tomato has been analyzed by microarray hybridization, 474 tomato transcripts were detected as putatively mobile mRNAs. Some of them were positively validated by RT-PCR and sequencing. These results suggested that a symplastic continuum exists between host and parasite that enables the macromolecular exchange. Interestingly, not all transcripts known to be phloem-mobile in the host plant were transferred to dodder. A surveillance control mechanism may exist at the interface between host and parasite to control the movement of potentially mobile transcripts. This raised the question of which organism—host or parasite—regulates movement between the species. A practical aspect of RNA movement between the host plant and parasite could be controlling the plant parasitism via RNA interference. Indeed, a target transgene expression in the hemiparasite *Triphysaria versicolor* was shown to be silenced by RNAi expressed from a transgene in the host plant lettuce, though the level of silencing appeared to be highly variable in different host–parasite pairs [63]. Furthermore, when non-transgenic *Triphysaria* seedlings were used as bridges connecting lettuce plants expressing GUS RNAi with those expressing GUS, a silencing effect was observed in the GUS-expressing lettuce plant in some of these double-junction associations. These results showed that reciprocal movement of RNAi from parasite to host also occurred. Similar results were obtained when *Triphysaria* linked the host plant of two different species, the GUS RNAi expressing lettuce and GUS-expressing *A. thaliana*. Over half of the associations showed a silencing effect in both cases. Thus, the silencing signal had moved with an equal efficiency between hosts of different families. Notably, the degree of silencing was negatively correlated with the length of *Triphysaria* root between the two host plants. In a study, the potential of transferred RNAi was tested for silencing dodder genes involved in haustorial development [64]. The dodder *KNOX1* family genes *SHOOT MERISTEMLESS-like* (*STM*) and *KNOTTED-LIKE FROM ARABIDOPSIS THALIANA1-like* (*KNAT1-like*) were used as target genes. To examine the function of these genes in dodder parasitism, transgenic tobacco plants were created that expressed an RNAi construct targeting the 3′-untranslated regions of both dodder *KNOX1* genes but did not share sequence similarity to the tobacco orthologs. By reasons that remained unknown, no silencing of the *KNAT1-like* gene was observed. No changes in the *STM* expression level were observed in dodder strands at the attachment points with mature haustoria that had low levels of *STM* expression both on transgenic and wt tobacco hosts. However, strong down-regulation of *STM* expression occurred in young developing dodder strands that showed the highest *STM* expression on the wt hosts. Dodder grown on *KNOX1* RNAi transgenic hosts showed poor establishment and decreased growth when compared with those grown on the wt plants. These results show that *STM* expression might be important at earlier steps of haustorial development.

Transcriptome sequencing was used to investigate the RNA transfer between *C. campestris* and its hosts, genome-wide [65]. To this end, *C. campestris* plants were grown on *A. thaliana* and tomato hosts and three tissue samples were harvested from each parasite–host association, namely the host stem above the region of attachment (HS), interface region where parasite was connected to the host (I), and the parasite stem near the region of attachment (PS). In *Arabidopsis*-*Cuscuta* associations, ~1.1% of *A. thaliana* reads were detected in the PS samples, while the HS samples contained ~0.6% of *Cuscuta* reads. About equal numbers of *Cuscuta* and *Arabidopsis* reads were detected in the I samples. In tomato-*Cuscuta* associations, ~0.17% of tomato reads were detected in the PS samples, and ~0.38% of *Cuscuta* reads were detected in the HS samples, suggesting a lower rate of transfer than in *Arabidopsis*-*Cuscuta*. In the I samples, ~86% reads were from tomato, probably due to a greater mass of the tomato stem tissue. Thus, *Cuscuta* is capable to transfer mRNAs between different plants. The host mRNAs in the parasite stem occurred in the fully spliced form. The greatest number of mobile transcripts was from *Arabidopsis* to *Cuscuta;* ~45% (9518) of the expressed *Arabidopsis* transcripts were found in *Cuscuta*. In contrast, only 1.6% (347) of the expressed tomato transcripts showed strong evidence of mobility. With respect to movement from parasite to host, ~24% (8655) of *Cuscuta* transcipts were detected in *Arabidopsis* stem, while only ~0.8% (288) of *Cuscuta* transcripts were detected in tomato stem. Thus, a much larger exchange occurred between *Cuscuta* and *Arabidopsis* than between *Cuscuta* and tomato. Probably, some host-specific mechanism controls haustorial selectivity of mobile mRNAs. One feature that seemed to be essential for this selectivity was the relative abundance at the parasite–host interface. However, many transcripts with similar abundances had different mobilities. Therefore, it was not the only factor affecting mobility. GO analyses of mobile and not mobile mRNAs in *Arabidopsis* and *Cuscuta* revealed some differences in mobility between gene functional groups. Thus, a high tendency for being mobile was found in the “response-to-stimulus” genes, suggesting a possible specific targeting for mobility, but the mechanistic basis for such specificity remained obscure. An intriguing possibility that the m^5^C-methylation may target mRNAs for inter-plant mobility over graft junctions was recently described [66]. Levels of most *Arabidopsis* mobile transcripts in the *Cuscuta* stem were by two orders of magnitude lower than their levels at the *Arabidopsis*–*Cuscuta* interface. However, some host RNAs appear to move much more readily as evidenced by their about equal levels between these two compartments. The ability of one *Cuscuta* plant to bridge different host individuals raises the possibility that this parasite could mediate RNA exchange across individuals of different species.

When the *N. benthamiana* plants were infected with a TRV vector containing fused fragments of three broomrape *P. aegyptiaca* genes essential for the parasite life cycle and then challenged by the parasite infestation, significant down-regulation of the target genes in the parasite tissue was observed, and the number and size of the parasite tubercles was reduced [67]. Similar effects were observed when the transgenic tomato plants expressing the same fused fragments of target genes in a hairpin configuration were used as the host plants. In both models, the silencing of the target genes in the parasite correlated with the expression levels of respective siRNAs in the host. Thus, siRNAs produced in the host plants could be effective in controlling the infectivity of parasite plants via silencing the parasite genes essential in the infestation process.

Since the movement of RNA molecules through the *Cuscuta* haustoria was shown to be bidirectional, it could be easily conceived that the interspecies siRNA silencing effects between the host and parasite might be mutual. Indeed, in the *A. thaliana*–*C. campestris* associations, the host plant genes were shown to be silenced by the parasite-produced mobile siRNAs [26]. Very similar to a previous paper of the same authors [65], sRNA expression was profiled in three tissue samples from parasite–host associations: the host stem above the region of attachment (HS), interface region between the parasite and the host (I), and the parasite stem near the region of attachment (PS). As expected, levels of most *C. campestris* sRNAs in the sample I were lower than in the sample PS. Unexpectedly, 76 sRNA species of *C. campestris* were found to be significantly up-regulated in the interface region relative to the parasite stem. Of these interface-enriched species, 43 were canonical miRNAs. One of these 43 miRNAs was a member of the conserved miR164 family, while others had low sequence similarity to known *miR* loci, and none of them aligned perfectly with the *A. thaliana* genome. Of the 43 up-regulated miRNAs, the majority (26) were 22-nt miRNAs known to be associated with the accumulation of secondary siRNAs from their targets to amplify miRNA-directed gene silencing. Six *A. thaliana* mRNAs were found to be potentially targeted by *Arabidopsis* siRNAs that were preferentially accumulated in the interface region and showed complementarity to the *Cuscuta* up-regulated miRNAs. These mRNAs encoded partially redundant auxin receptors TIR1, AFB2, and AFB3; a membrane-localized kinase BIK1 required for pathogen and developmental signaling; an abundant phloem protein SEOR1 that reduces photosynthate loss from the phloem after injury; and a transcriptional repressor HSFB4 required for the formation of the root stem cells. *TIR1*, *AFB2*, and *AFB3* were known to be targeted by the 22-nt miRNA393 and to produce secondary siRNAs downstream of the miRNA393 target sequence. However, in parasitized stems, the location of such secondary siRNAs was in a more upstream region proximal to the sites of complementary to the *C. campestris* miRNAs. Therefore, these secondary siRNAs were likely triggered by the *C. campestris* miRNAs, not by the host miRNA393. None of the *A. thaliana* miRNAs or siRNAs that could target these six mRNAs was found to be induced in *Arabidopsis–Cuscuta* interaction. Likewise, none *C. campestris* secondary siRNAs corresponding to the induced miRNAs were found. Some dodder orthologues of *TIR1*, *HSFB4*, and *BIK1* had poor complementarity to these miRNAs. Hence, the induced *C. campestris* miRNAs might evolve to avoid targeting the *Cuscuta* own transcripts, but act in a trans-species manner to down-regulate *A. thaliana* mRNAs. Consistent with this view, levels of five of these six target mRNAs was significantly reduced in parasitized compared with control stems. Furthermore, accumulation of the *A. thaliana* secondary siRNAs suggests that *Cuscuta* hijacks the host’s silencing machinery to amplify the effects of its host-targeting miRNAs. When *Arabidopsis* plants null-mutant for either of the two putative targets of induced miRNAs, *SEOR1* and *AFB3*, were used as the host plants, significantly larger biomass of *C. campestris* was observed. Therefore, both these genes restrict the *C. campestris* growth, and their down-regulation by parasite-produced mobile miRNAs is biologically relevant. Orthologues of *BIK1*, *SEOR1*, *TIR1,* and *HSFB4* as putative targets of the interface-induced *Cuscuta* miRNAs were found in many eudicot species. Overall, these data show that the interspecies siRNA-induced silencing evolved as a double-edged weapon in the “arms race” between the host and parasite plants. An analysis of mobile sRNAs in four *Cuscuta* species showed that these sRNAs are mostly unique to each species [68]. Altogether, 61 target sites over 54 *A. thaliana* mRNAs were found for mobile sRNAs from the four *Cuscuta* species. Accumulation of these target mRNAs was shown to be down-regulated in parasitized host stems. Thus, similar to *C. campestris* [26], other *Cuscuta* species also use trans-species sRNAs that affect gene expression in their host plants. In some cases, mobile sRNAs from multiple *Cuscuta* species were found to target the same *Arabidopsis* mRNAs, such as *SEOR1*. However, the majority of mRNAs were uniquely targeted by sRNA of single *Cuscuta* species. It has been suggested that these trans-species sRNAs in different *Cuscuta* species regulate similar host processes, but not necessarily via the same mRNAs. By the sequence similarity clustering, many superfamilies of *Cuscuta* mobile sRNAs were found, including multiple superfamilies that were shared between different isolates and species of *Cuscuta*. About half of them were miRNAs. In many cases variation within superfamilies occurred in a visible three-nucleotide period, suggesting targeting similar mRNAs with synonymous codon variation. Importantly, superfamily variation occurred within single *Cuscuta* species, such that multiple variants of the same sRNA superfamily were commonly deployed by a given parasite during an infestation. In such a way, targeting of conserved sites in orthologous mRNAs of various host plant species could be achieved. Using sRNA-seq libraries made from *C. campestris* attachments on *Arabidopsis* and tobacco, this view was directly confirmed. The *N. benthamiana TIR1* target sites appeared to encode identical amino acids, but vary at synonymous positions, and some variants of the *C. campestris* sRNA superfamily SupFam_27 showed more complementary to either tobacco or *Arabidopsis TIR1* target sequence. sRNA superfamily diversity could also enable the repression of multiple mRNAs within a single host. In *Arabidopsis*, ten gene families were found to be putative targets of multiple *C. campestris* mobile sRNAs, including WRKY motifs within a family of defense-related TFs. Targeting conserved protein-coding sites by sRNA variants that cover most possible synonymous site variations appears to be a *Cuscuta* strategy to cope with the evolution of host resistance by target site sequence changes, and to interact with diverse hosts.

In the *A. thaliana*–*C. campestris* associations, enrichment for 22-nt siRNAs was observed in parasitized host stems relative to non-parasitized host stems, which were enriched in 21-nt siRNAs [69]. No significant enrichment was observed for any length of sRNAs in *C. campestris*. Concomitant with the enrichment of 22-nt siRNAs, up-regulation of the two genes involved in secondary siRNA biogenesis, *SGS3* and *RDR6*, was observed in the apical region of parasitized host stems, indicating that the production of secondary siRNAs in the host was activated by parasitism. The mobility in the recipient stem was tested for putative mobile sRNAs that were abundant in the interface region and could be unequivocally identified as originating from either *Arabidopsis* or *C. campestris*. Of the seven candidate *C. campestris*-derived sRNAs tested, two were confirmed to move into apical regions of the parasitized *Arabidopsis* stems. Likewise, of the six candidate *A. thaliana*-derived sRNAs tested, two were confirmed to move into apical region of parasitizing *C. campestris* stems. Notably, one of the *Cuscuta*-derived mobile sRNA moved to the distant organs of the *Arabidopsis* host plants, while the other one was detected in the interface region only. Thus, mobility in the recipient tissues differs between different trans-species sRNAs. Predicted targets of the highly mobile *Cuscuta*-derived sRNA in the host cells included disease resistance protein family genes and sRNA degrading nuclease gene, while those of the less mobile sRNA included Golgi nucleotide sugar transporter gene and the *ARF-GAP domain 6* gene. Predicted targets of the *Arabidopsis*-derived sRNAs in the *Cuscuta* cells included a metallopeptidase M24 family protein gene, an RNA-binding protein gene, and several LRR protein kinase family protein genes.

It was found, by proteomic analyses, that hundreds of proteins are translocated between the stems of the dodder and those of its host plants [70]. Moreover, the exchange of hundreds of proteins occurred between the two host plants of different species bridged by a dodder parasite. The majority of these proteins were bona fide mobile proteins, not translated from the mobile mRNAs. About 27.0% (734/2714) of the total *Arabidopsis* proteins and 37.9% (1807/4765) of the total soybean proteins identified in their host stems were detected in dodders, while 17.4% (1027/5914) and 28.3% (1674/5914) of the total dodder proteins were detected in *Arabidopsis* and soybean stems, respectively. Among these mobile proteins, six *Arabidopsis* and eight soybean TFs were translocated to dodder, while four and nine dodder TFs were translocated to *Arabidopsis* and soybean, respectively. Some pathogen-resistance and insect-resistance proteins were also translocated that may confer resistance to respective biotic stresses in companion plants. When the *Arabidopsis* and soybean hosts were bridge-connected by a dodder plant, 20.1% (949 of 4716) of *Arabidopsis* proteins and 11.6% (719 of 6193) of soybean proteins were translocated to the other host. Therefore, dodder mediates extensive protein trafficking between different host plants. Notably, among the proteins transferred between the two hosts, 217 were orthologous, while 657 and 462 were specific for *Arabidopsis* and soybean, respectively. And again, some TFs and resistance proteins were among the transferred.

A trans-species transfer of long non-coding RNAs (lncRNAs) between the dodder and its host plant soybean has been detected in a recent study [71]. In *C. australis* stems, 14 lncRNAs were identified as originating from the soybean, while in the soybean stem, 365 lncRNAs were from *Cuscuta*. Apparently, bidirectional movement of lncRNAs occurred between the dodder and soybean. Likewise, 74 mRNAs in *Cuscuta* stems were from soybean, and 8894 mRNAs in soybean stems were from *Cuscuta*. Therefore, bidirectional movement of both lncRNAs and mRNAs occurred between the host and parasite plants, but the movement from the parasite to the host was strongly predominant.

Collectively, these findings demonstrated a wide scope of the molecular exchanges between parasite and host plants. The functional significance of these exchanges mostly remains obscure. However, in some cases, the macromolecules transferred appear to function in the companion plant. Thus, *Cuscuta australis* lacks the gene for the functional FLOWERING LOCUS T (FT) regulator of flowering and appears to use the host-derived FT protein to synchronize its flowering with the flowering of the host [72]. The inter-species trafficking of the phosphinothricin acetyl transferase from the parasitized soybean plants endows the parasitizing dodders with the glufosinate herbicide tolerance [73]. Likewise, glucosinolates in *Arabidopsis* can be translocated to *Cuscuta gronovii* and protect this parasite against pea aphids (*Acyrthosiphon pisum*) [74].

### 4.5. Interplant Signaling

It was demonstrated that in *Cuscuta*-connected plant clusters, *Cuscuta* mediates herbivory-induced interplant signaling that activates defense responses and primes the undamaged host plants in *Cuscuta* bridge-connected hosts against subsequent insect attack [75]. Soybean plants were infested with *C. australis* vines, and, during the vigorous growth stage, pairs of soybean plants were created by placing two *C. australis*-infected soybean hosts next to each other and allowing *C. australis* vines to parasitize and connect the two hosts. Then soybean leaves (local leaves; named L leaves) were infested with *Spodoptera litura* larvae or left untreated (controls). After 24 h of feeding, the L leaves, the systemic leaves of the caterpillar-attacked plants (S1 leaves), the leaves of the connected but undamaged plant (S2 leaves), and the vines of *C. australis* were harvested for RNA-seq analysis. Compared with the control group, 820 genes were up-regulated and 84 genes down-regulated at least twofold in the L leaves after *S. litura* infestation, and 519 genes up-regulated and 136 genes down-regulated in the S1 leaves, demonstrating the spreading of defense signals within the attacked plant. Intriguingly, 283 genes were up-regulated and 283 genes down-regulated in S2 leaves, suggesting that herbivory-induced signals are transferred via the *C. australis* bridge to the second host plant. Most DEGs were specifically regulated in L (703), S1 (362), or S2 (353) leaves. Only 49 genes were commonly regulated, among which seven showed different directions of expression changes between L, S1, or S2 leaves. Two genes, *Glyma09g28310* and *Glyma16g33710*, which encode important anti-insect proteins–trypsin protease inhibitors (TPIs), were up-regulated in all L, S1, and S2 leaves, suggesting that in response to herbivory all *Cuscuta*-connected soybean plants may have elevated defense levels. After 48 h of *S. litura* feeding on L leaves, the TPI activity in the S2 leaves was by ∼40% greater than in the control group, and these S2 leaves consistently exhibited elevated resistance to insects. Given that hundreds of DEGs were detected in S2 leaves, it is likely that other induced defenses, in addition to TPIs, also contributed to this elevated resistance. The RNA-seq analysis of *C. australis* vines revealed 140 differentially expressed genes (DEGs) (79 and 61 up- and down-regulated, respectively) after *S. litura* feeding on L leaves, including genes encoding LRR proteins, cytochrome P450s, Ca^2+^-binding proteins, and proteins from the superfamily of protein kinases. The most significantly altered pathways were all related to primary metabolism. Similar transfer of insect resistance via *Cuscuta* bridges was also observed in heterospecific hosts systems, which consisted of *Arabidopsis*–*C. australis*–tobacco or soybean–*C. australis*–wild tomato combinations [75]. To investigate the role of JA in *Cuscuta*-mediated systemic signaling between plants, wild-type (wt) and *dde2-2 Arabidopsis*, which harbors a mutation in the JA biosynthesis *AOS* (*allene oxide synthase*) gene, were connected pairwise to tobacco plants. *Arabidopsis* plants were untreated or squeeze-damaged with a pair of forceps, and, after 8 h, the leaves of the tobacco were harvested for RNA-seq analysis. Compared with the untreated control group, 1342 DEGs (981 and 361 up- and down-regulated, respectively) were identified in the tobacco of the wt wound-treatment group, and the most strongly enriched pathways included “13-lipoxygenase (13-LOX) and 13-hydroperoxide lyase (13-HPL)” and “JA biosynthesis”. In contrast, wounding *dde2-2* plants resulted in only 404 DEGs (191 up- and 213 down-regulated) in the tobacco, and “nitrate reduction II (assimilatory),” “ammonia assimilation cycle II,” and “glutamine biosynthesis I” were among the enriched pathways. Notably, two genes encoding TPIs, *SS15144g00001* and *SS72472g00002*, were up-regulated 18- and 2-fold, respectively, in the tobacco connected with wt *Arabidopsis*. The expression level of the third TPI gene, *SS2003g01003*, was not detectable in controls but relatively high after induction. Only *SS15144g00001* was induced 3-fold in the tobacco plants connected with *dde2-2 Arabidopsis*. Overall, 240 tobacco genes were commonly regulated in the tobacco plants in both wt–*Cuscuta*–tobacco and *dde2-2*–*Cuscuta*–tobacco plant clusters. Probably, their transcriptional regulation does not depend or is partially dependent on the JA pathway. Thus, the *Cuscuta* bridge connections transmit conserved insect herbivory-induced systemic signals among different conspecific or heterospecific hosts, and, importantly, this interplant signaling plays an important role in priming all connected hosts in clusters against insect attack. Furthermore, the JA pathway in the local wounded plants is crucial in initiating and/or maintaining the systemic signals, and it is possible that there is more than one mobile signal that could travel through *Cuscuta* bridges between plants. In *Arabidopsis*–*C. australis*–*Arabidopsis* plant clusters, when *Arabidopsis* 1 was wounded to mimic insect feeding, transcriptional responses in *Arabidopsis* 2 were detected as early as 30 min later, when the transcript abundances of three wound-inducible systemic marker genes, *ZAT12*, *WRKY40*, and *WRKY53*, increased several folds. Similarly, in heterospecific tobacco–*C. australis*–*Arabidopsis* plant clusters, 10-fold increased levels of *WRKY40* in the untreated *Arabidopsis* were observed at 45 min after tobacco wounding. Thus, the systemic signal travels at a relatively high speed. In a chain of six *Arabidopsis* plants (A1–A6) connected in a row by five individual *C. australis* plants, the wound signal transfer along the whole chain (∼100-cm total distance) was observed. After the first plant (A1) was wounded, the transcript levels of *ZAT12*, *WRKY40*, and *WRKY53* in A2–A6 were determined. At 90 min, these genes’ transcript levels in the A4 plant were similar to those in A2 plant at the 45-min time point, while the transcript levels in A2 were already decreasing at 90 min. These results suggest that the wound response was propagated as a wave through the connected plants. As the A6 plant also showed detectable responses, it could be concluded that the *Cuscuta*-mediated interplant signaling can occur along multiple hosts reaching at least ∼100 cm. To examine whether this interplant signaling can activate defenses in the distal plants, the A6 *Arabidopsis* plant was replaced with a tobacco plant. The *Arabidopsis* A1 was infested with *S. litura*, and, after 48 h, tobacco leaves were harvested for determination of TPI activity or infested with *S. litura* for another 72 h. The TPI activity in these tobacco plants was more than twofold greater than in the non-treated control group, and these plants consistently exhibited elevated resistance against *S. litura*. Thus, the *Cuscuta*-mediated interplant systemic signaling occurs rapidly and can induce defense responses even in distantly connected hosts. The nature of the mobile signals and their means of propagation remained mostly unknown, though JA signaling probably plays a role. To test the involvement of JA signals, tobacco plants in which JA biosynthesis was silenced by the *allene oxide cyclase* (*AOC*)-specific RNAi were used as the hosts, and the responses of dodders and the host plants to herbivory by *S. litura* caterpillars on the dodders were investigated [76]. After the caterpillar attack, 162 DEGs were identified in the dodders grown on wt tobacco, more than 90% of them up-regulated. In contrast, in dodders parasitizing AOC-RNAi tobacco, only 56 DEGs were detected, and these were all up-regulated. Only 33 DEGs were common between dodders grown on wt and AOC-RNAi tobacco. Among the 162 DEGs obtained from the dodders grown on wt tobacco, the terms “response to chitin”, “response to wounding”, “response to water deprivation”, “response to JA stimulus”, and “JA mediated signaling pathway” were enriched, while the GO terms “response to chitin”, “response to ABA stimulus”, “ethylene mediated signaling pathway”, “response to wounding”, and “defense response to bacterium” were enriched among the 56 DEGs from the dodder grown on AOC-RNAi tobacco. Unexpectedly, 351 DEGs were found between the dodders growing on AOC-RNAi and wt tobacco even without the caterpillar attack, most of them (286) down-regulated in AOC-RNAi. Thus, the host JA signaling plays a role in regulating the gene expression in the parasitizing dodders both in the normal and insect-attack situations. In wt tobacco 2429 and 650 genes were up-regulated and down-regulated, respectively, at 12 h of caterpillar feeding. In AOC-RNAi tobacco, 1408 and 668 genes were up-regulated and down-regulated, respectively. A dendrogram analysis showed that in the absence of herbivory wt and AOC-RNAi tobacco plants had similar transcriptomes, while after the herbivory treatment of their respective parasitizing dodders, the wt transcriptome still grouped with those of un-attacked wt and AOC-RNAi plants, while the AOC-RNAi transcriptome became remote from the three others. Consistent with these results, only 485 DEGs were commonly regulated between the herbivory-treated wt and AOC-RNAi host plants. Thus, caterpillar feeding on dodder induced large transcriptomic changes in the host plant, and these changes were largely dependent on the host JA pathway.

Unlike caterpillars that damage large areas of tissue, leading to a rapid increase in the concentration of JA and changes in other defense-related hormones, aphids are piercing–sucking feeders that do not cause large lacerations and tissue losses. In a recent study, local defense responses in the *C. australis* attacked by the green peach aphid (GPA) have been investigated, as well as systemic signals from the attacked parasite to the host plants [77]. The *Cuscuta*-soybean associations were grown for 4 weeks before the GPA treatment of the *Cuscuta* exploratory stems at a position ~10 cm away from haustorial insertions. After 24 h of GPA feeding, *Cuscuta* and soybean tissues were harvested and used for the phytohormone determination and RNA-seq analysis. Compared with the control plants, the SA and JA contents in GPA-treated *C. australis* decreased by 58% and 41%, respectively, while in the host soybean plants that were not attacked by GPA, the contents of JA increased by ~3.4-fold, whereas the SA levels were unchanged. Compared with control plants, GPA attacked dodders showed 172 DEGs, 90 and 82 of them up- and down-regulated, respectively. Strikingly, many more DEGs were identified in the soybean plants. Unlike dodders that showed about equal numbers of up- and down-regulated genes, 324 genes were up-regulated and 691 genes down-regulated in soybean. The most significantly altered pathways were “JA biosynthesis”, “divinyl ether biosynthesis II”, “simple coumarins biosynthesis”, “cellulose biosynthesis”, and “phenylpropanoid biosynthesis”. Fourteen TFs were induced or repressed in attacked dodders, belonging to eight families: MIKC, bHLH, HB-other, HSF, Trihelix, ERF, bZIP, and WRKY. Among these TFs, five were up-regulated and nine down-regulated. In the soybean, 108 TFs from 24 families were regulated; half of them belong to the ERF, bHLH, NAC, MYB, and MYB-related TF families. Among these TFs, 20 were up-regulated and 88 down-regulated. Notably, the top 15 most regulated TFs were all down-regulated, including seven ERFs. These wide changes of gene expression in soybean were found to increase their resistance to the cotton leafworm larvae and to the soybean aphids. In an opposite experimental setup, feeding of leafworm larvae or soybean aphids on the soybean host plants did not change the *C. australis* resistance to GPAs. This finding was somewhat unexpected, since feeding of leafworm larvae on soybean hosts have been shown to change gene expression in parasitizing *Cuscuta* [75]. The inter-plant systemic signals between parasites and their host plants may provide advantage to host plants in some aspects, though overall the host fitness is decreased by parasitism. These findings were recently further advanced to show that in a GPA–*C. australis*–cucumber association, in which GPAs fed on *C. australis* that was parasitizing on the cucumber, not only GPAs feeding induced defense-related responses in *C. australis* and cucumber, but also large numbers of mRNAs were found to be transferred between *C. australis* and cucumber and between *C. australis* and GPAs [78]. Even more unexpectedly, it was also found that a few mRNAs from GPAs were detected in cucumber and some mRNA from cucumber were detected in GPAs, indicating extensive cross-species mRNA trafficking.

A recent work from Jianqiang Wu laboratory appears to be the only study where both transcriptome and DNA methylome changes were investigated in *C. campestris*-connected plant clusters [79]. The system was used to investigate whether different host plants are able to communicate systemic signals of N starvation to affect each other’s transcriptome and DNA methylome. Transcriptomes were compared by RNA-seq between host plant pairs consisting of two N-repleted hosts (CN+), two N-depleted hosts (CN-), and one N-depleted (TN-) and one N-repleted (TN+) host. The transcriptome data from CN+ roots and leaves were used as a baseline to determine the DEGs in the roots and leaves of CN-, TN+, and TN- plants. When two cucumber plants were used in the pairs of *C. campestris*-connected host plants, in the CN- plants, more than 3000 and 1200 DEGs were detected in roots and leaves, respectively, while in the roots and leaves of TN- plants, at most 2136 and 1024 DEGs were found. Strikingly, even though TN+ plants were not treated with N starvation, their roots and leaves exhibited large numbers of DEGs, more than those induced in the TN- and CN- plants. The transcriptome profiles of CN- and TN+ leaves showed high similarities. In roots, the transcriptomes of TN+ plants were similar to those of the CN- and TN- plants at 2 and 8 h, while at 48 h, TN+ and CN- transcriptomes were rather different. These data suggest that the N-depleted TN- plants constantly sent systemic signals to the N replete TN+ plants, leading to dramatically altered transcriptomes in both TN+ leaves and roots, and importantly, certain mobile signals from the TN+ plants were also transferred to the TN- plants and induced changes in their gene expression. Since both TN+ and CN+ plants were supplied with N, their transcriptomic differences were caused by systemic signals transferred to TN+ plant from the TN- plants via the *C. campestris* bridge. Likewise, the differences between the transcriptomes of TN- and CN- plants revealed the effect of the systemic signals from the TN+ plants. At most, 809 DEGs were found between TN- and CN-, while many more (up to 3514) were found between TN+ and CN+, in both leaves and roots. At most 809 and 16 genes in the leaves and roots, respectively, of TN- plants were specifically regulated by the systemic signals from TN+ plants at 8 h, while at most 1383 (48 h) and 3514 (2 h) in the leaves and roots, respectively, of TN+ plants were specifically regulated by the systemic signals from TN- plants. Of these cross-regulated genes, 180 were common in the leaves, implying that the respective systemic signals were bi-directionally transferred between the TN- and TN+ plants, though at different rates. In contrast, only six genes were common in the roots. GO enrichment analysis indicated that “phosphorus metabolic process”, “organonitrogen compound metabolic process”, “protein phosphorylation”, and “anion transport” were enriched in the DEGs from TN+/CN+. Nine GO terms were enriched in TN-/CN-, including “response to hormone”, “response to auxin”, and four terms related to photosynthesis. These data support the scenario that N-starvation-induced systemic signals moved from TN- to TN+ plants through *C. campestris* bridges, and also that certain systemic signals originating in TN+ traveled to the TN- plants, leading to differences in gene expression between the TN- and CN- plants. Therefore, *C. campestris* is able to transfer certain systemic signals bi-directionally between the N-depleted and N-replete hosts, and these systemic signals have strong regulatory effects on the transcriptomes of the recipient host plants. Relative to the numbers of DEGs in host plants, *C. campestris* exhibited almost negligible numbers (2–10% of the numbers in the hosts), suggesting that *C. campestris* can transmit the signals, but is not transcriptionally sensitive to them.

Under unstressed conditions, the transcriptomes of CN+ roots and leaves contained only 8 and 60 *C. campestris* mRNAs, respectively, whereas 1025 and 789 *C. campestris* mRNAs were detected at 2 h in the CN- cucumber leaves and roots, respectively, and the numbers of *C. campestris* mRNAs transported to CN- plants dropped sharply over time (fewer than 10 in both roots and leaves at 48 h). However, the numbers of mRNA species translocated from CN- hosts to *C. campestris* had increased to 1653, 3878, and 1805 at 2, 8, and 48 h, respectively. Thus, N stress elevated the degree of inter-plant mRNA translocation between *C. campestris* and host. In the TN+~TN- clusters, the cucumber mRNAs in *C. campestris* decreased from 725 at 2 h to 252 at 48 h. Strikingly, while the TN+ roots resembled the CN+ roots, having at most 11 *C. campestris* mRNAs at all times, the TN- roots were found to have 315, 1793, and 922 *C. campestris* mRNAs at 2, 8, and 48 h, respectively. In the TN- and TN+ leaves, at most 825 and 1320 *C. campestris* mRNAs were identified, and the numbers of the inter-plant mobile *C. campestris* mRNAs had decreased to 3 and 102 in TN- and TN+ leaves, respectively, at 48 h. Therefore, under heterogeneous N conditions, communications among N-replete and N-depleted hosts and *C. campestris* have a strong impact on inter-plant mobile mRNAs.

To investigate whether the hosts in plant clusters may respond to N-systemic signals with DNA methylation changes, the samples collected at 48 h were used for methylome analysis. The methylation patterns of CN+ roots and leaves were used as baselines to identify the differentially methylated regions (DMRs) in TN+, TN-, and CN- roots and leaves. In total, 4119, 1482, and 1335 DMRs were identified in TN+, TN-, and CN- plant leaves, respectively, and 1780, 3092, and 18,413 DMRs were found in the roots. The CHH sites and CHG sites showed the highest numbers of DMRs in roots and in leaves, respectively. The dendrograms based on the three types of cytosine methylation (CG, CHG, and CHH) sites showed that the leaf methylomes of TN+ and TN- plants were the most similar, while the methylome of CN- plants was the most dissimilar to those of other plants. Not only CN- and TN- plants responded to N starvation with changes to the methylome, but the DNA methylation patterns of TN+ plants also revealed large changes, likely due to the influence of systemic signals from the TN- plants. Compared with TN+ and TN- plants, CN- plants exhibited the largest number of DMRs, and in particular, the roots of CN- plants had the largest numbers of differentially methylated CG, CHG, and CHH sites. Importantly, in both leaves and roots of TN+, TN-, and CN- plants, many specific sites with CG, CHG, or CHH methylation changes were identified. Thus, N starvation induced large-scale methylome alteration not only in the N-depleted CN- and TN- host plants, but also in the N-replete TN+ plants. Moreover, TN+ and TN- plants had unique DNA methylation changes, likely due to the bilateral systemic signaling.

Next, the DMRs between TN+ and CN+ plants and between TN- and CN- plants were inspected to reveal the differential DNA methylation patterns induced by the systemic signaling crosstalk between TN+ and TN- plants. Again, many specific methylation changes in CG, CHG, and CHH sites were detected in TN+/CN+ and in TN-/CN- comparisons, suggesting that the N-depleted TN- plants and the N-replete TN+ both sent systemic signals to regulate each other’s DNA methylation. Importantly, the methylome data from the TN+ and TN- hosts supports the notion that in addition to the N-deficiency-induced systemic N signals from the TN- to TN+ plants, certain systemic signals from the TN+ plants also travel to TN-, and these different systemic signals from the TN+ and TN- plants crosstalk to activate specific changes of transcriptomes and DNA methylomes in the TN- and TN+ plants.

In similar experimental setups, the transfer of the salt stress signals between pairs of the cucumber [80] and sweet orange [81] plants via dodder bridges were studied by transcriptomic analyses. At 1 h after the salt treatment, 363 DEGs were identified in non-treated companion cucumber plants relative to controls, indicating a rapid transfer of systemic signals from salt-stressed plants [80]. Notably, hormone signaling-related genes were among the top 40 most regulated DEGs, implying that phytohormones may be involved in signaling regulating this early systemic response to salt stress. A total of 1472 (472 up-regulated, 1000 down-regulated) and 1904 (981 up-regulated, 923 down-regulated) DEGs were identified at 12 h and 24 h in the salt treated orange plants [81]. Smaller transcriptome changes were observed in bridged non-stressed orange plants; 557 (74 up-regulated, 483 down-regulated) and 59 (19 up-regulated, 40 downregulated) genes changed expression at 12 h and 24 h, respectively. A total of 424 DEGs were common between the salt-treated and non-stressed plants at 12 h, indicating a high correlation between their responses to the treatment. Interestingly, a 24 h, the salt-treated plants showed more obvious salt stress-related patterns of gene expression, while their non-stressed bridged partners showed obvious transcriptional features of adaptation to salt stress. Thus, the systemic signals were transmitted via the dodder bridge from the salt-treated host plant to prime the companion non-stressed host plant to a possible incoming salt stress.

## 5. Conclusions

RNA sequencing analyses at the whole genome scale were widely used in the last years to identify genes potentially involved in the parasite–host plant interaction. Some of these genes appeared to be expressed at the parasitic stage in haustoria in various parasitic species. Thus, differential expression of auxin-related genes has been reported in many parasitic species. Likewise, genes encoding cell wall modification enzymes have been shown to be up-regulated in many parasitic species during host invasion as well as in many host species as a part of resistance response. Two lines of defense against different pathogens, pathogen-triggered immunity (PTI) and effector-triggered immunity (ETI), apparently have been coopted to act against different lines of parasite plants. It has been found that host plants can induce gene silencing in the parasite plants via mobile RNAi–host-induced gene silencing (HIGS)—a promising approach to the parasite plant control.

Epigenetic regulation is probably involved in adapting the parasite plants to different host plants. The existence of miRNA superfamilies that could potentially target conservative mRNAs in different host species is a good illustration. Other epigenetic mechanisms, DNA methylation in the first place, are probably also play important roles, similar to those in the plant adaptation to abiotic stresses and microbes. In the dodder–alfalfa interaction, changes in global levels of DNA methylation both in the parasite and the host plant tissues were described by one of us more than 40 years ago [82]. Nevertheless, the current knowledge concerning the DNA methylation and histone modifications in parasitic plants is scarce. Whether parasite plants use any kinds of epigenetic memory to improve adaptation to the host plants that they repeatedly parasitize is still mostly unknown. Likewise, we have very poor knowledge concerning the role of epigenetic mechanisms in the host plant responses to the parasitic plant attacks.

In the host-parasitic plant associations, exchange of various kinds of molecules between host and parasite takes place. In *Cuscuta* associations with different host plants, the exchange of regulatory signals occurs between hosts and parasite as well as between different hosts. Systemic signaling from the host that was attacked by a pest or other stress factor could increase the resistance of parasite to the same agent. On the other hand, the same signals could be transferred to non-attacked host plants via *Cuscuta* bridges to prepare them to better resist future attack. Thus, in many ways, this exchange could be beneficial for both the hosts and parasite, though, of course, the net effect of parasitism on host plants is usually negative. Though we know much about specific sets of mobile molecules, including mRNAs, small RNAs, and proteins, the regulatory mechanisms behind their selection remain unknown.

Massive gene losses and neo-functionalization of expanded gene families are the universal features in parasitic plants that might evolve to increase chances of their survival in the parasitic lifestyle. Horizontal gene transfer (HGT) is another characteristic feature in parasite plants [83]. The frequency of HGT events appears to be positively correlated with the parasite dependency on the host [84]. The number of HGT genes is maximal in obligate parasites, such as *Striga*, *Phelipanche,* and the endophytic holoparasite *Sapria*. The genome of holoparasitic *C. campestris* contains about 0.49% HGT genes, while in *Sapria* ~1.2% of uniquely aligned genome sequences were gained via HGT. Many of the shared HGT genes are highly expressed in haustoria, suggesting that they might evolve as important genes in plant parasitism.

There is a great need for the more exact knowledge about direct causal role of specific genes in the host plant resistance or susceptibility to parasitic plants and in the susceptible host selection by parasitic plants. Recently, gene knockdown by RNAi or gene over-expression from transgenes have confirmed the universal roles of hormone signaling and cell wall remodeling pathway, as well as of some specific TFs and receptor proteins in plant parasitism. Powerful CRISPR/Cas9 technology for selective gene knockout probably will be the tool of choice in such studies in the nearest future [85,86].

## Figures and Tables

**Figure 1 ijms-24-02647-f001:**
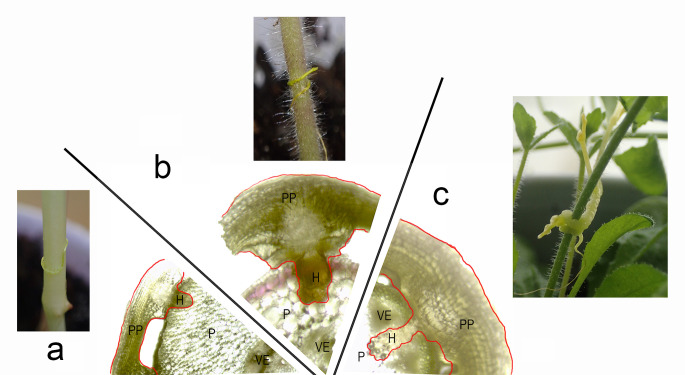
Differences in *Cuscuta campestris* haustoria development on different host species. The haustoria do not reach the vascular elements on corn (**a**) and shows a typical hypersensitive response on tomato (**b**), which lead to dying off of the parasite. A successful connection to the vascular elements is observed in *A. thaliana* (**c**), leading to successful secondary stem formation of the parasite. PP—parasitic plant; H—haustoria; P—parenchyma tissue; VE—vascular elements.

**Table 1 ijms-24-02647-t001:** Genome features of parasitic plants.

Species	Genome Size, Mbp; Gene Number	Gene Groups Lost	Gene Groups Gained	Reference; Sequencing Platforms; Assembly Metrics
*Striga asiatica*	472; 34,577	photosynthesis; leaf anatomy and function; abiotic and biotic stimuli responses	strigolactone perception	[10]; Ilumina+Sanger; N50 > 1.3 Mbp *
*Phtheirospermum japonicum*	1227; 30,337		subtilisin-like serine proteases	[13]; Illumina+PacBio; N50 > 1 Mbp
*Cuscuta campestris*	477; 44,303	photosynthesis; nutrient uptake from soil; RNA; stress; transport; lipid metabolism	DNA; protein	[14]; Illumina+PacBio; N50 > 1.38 Mbp
*Cuscuta australis*	273; 19,671	leaf and root development; nutrients uptake from soil; photosynthesis; flowering time; defense against pests and pathogens	response to hormones; DNA methylation; regulation of transcription; cell wall-related metabolism	[15]; Illumina+PacBio; N50 = 5.95 Mbp
*Sapria himalayana*	1280; 55,179	photosynthesis; defense; stress response; biosynthesis of ABA; protein degradation; purine metabolism; ubiquitin-proteasome-mediated protein degradation; endopeptidase Clp-mediated protein lysis	chromosome organization; DNA metabolism; cell cycle	[16]; Illumina+ONT; N50 = 4.3 Mbp

* 50% of the total assembly length covered by continuous scaffold sequences >1.3 Mbp.

## Data Availability

Not applicable.

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
