# Peer review of "Genomic and Epigenomic Mechanisms of the Interaction between Parasitic and Host Plants"

_ijms, 2023, doi:10.3390/ijms24032647_

Round 1

Reviewer 1 Report

Paper needs Major revision.

1- In Abstract, there is too lengthy introductory part. Reduce its length.

2- In Introduction, many statements are without any reference. Add suitable references to support these statement.

3- There is a need to improve/reformat some confusing sentences throughout.

4- Write full scientific name of an organism at its first citation in the text such as "S. hermonthica" on line 130 and "L. philippensis" on line 296.

5- Add appropriate references at various places to support the statements from Line 744 to 786.

6- There should be a uniformity in writing the scientific names of plants. At their first citation in the text, they should be written in full. However, in subsequent text, only abbreviated names of genera should be used followed by species name.

7- Format references uniformly and correctly.

8- In Table 1, format last column correctly. Write number of references. 

9- Do not cite Tables and Figures in Conclusion.

10- Conclusion is too lengthy. No need to cite references in this section.

11- Write scientific names in Italics especially in References.

Author Response

First of all, we want to thank the reviewer for the great job that he has done to improve our English. We accept all the suggested corrections and have made respective changes to the text. We have kept the full names of the plant genes in the Italics and the upper case in accordance with respective Guidelines (Bowman et al. The Naming of Names: Guidelines for Gene Nomenclature in Marchantia. Plant Cell Physiol. 2016, 57, 257–261, doi:10.1093/pcp/pcv193).

Following are our replies to the Reviewer’s specific comments and suggestions:

1 - In Abstract, there is too lengthy introductory part. Reduce its length.

Reply: We have removed a sentence from the introductory part. We think that other sentences are essential to substantiate our choice of the paper topic and should be kept.

2 - In Introduction, many statements are without any reference. Add suitable references to support these statements.

Reply: We have added the references.

3 - There is a need to improve/reformat some confusing sentences throughout.

Reply: We have reformatted all the sentences that were marked as confusing.

4 - Write full scientific name of an organism at its first citation in the text such as "S. hermonthica" on line 130 and "L. philippensis" on line 296.

Reply: “Striga hermonthica” was first cited by the full name above (line 116). The full name “Lindenbergia philippensis” has been added on line 296. All other names were carefully checked and corrected as needed.

5 - Add appropriate references at various places to support the statements from Line 744 to 786.

Reply: Significant part of the text indicated that described too much detail has been removed as suggested by other reviewers. The remaining text is supported by the reference [30] above.

6 - There should be a uniformity in writing the scientific names of plants. At their first citation in the text, they should be written in full. However, in subsequent text, only abbreviated names of genera should be used followed by species name.

Reply: We have made the respective corrections to the text.

7 - Format references uniformly and correctly.

Reply: Done.

8 - In Table 1, format last column correctly. Write number of references. 

Reply: Done.

9 - Do not cite Tables and Figures in Conclusion.

Reply: Tables and Figure and respective citations were moved to the main section.

10 - Conclusion is too lengthy. No need to cite references in this section.

Reply: We have removed from Conclusions the Figure and some references that were cited above, but kept the references that were specifically chosen to support our general conclusions on the subject.

11 - Write scientific names in Italics especially in References.

Reply: Done.

Reviewer 2 Report

Reviewer's report 

Date: 6 January, 2023 

Journal: International journal of molecular sciences

Manuscript ID: ijms-2165049

Type of manuscript: Review

Title: Genomic and epigenomic mechanisms of the interaction between parasitic and host plants

Authors: Vasily V. Ashapkin *, Lyudmila I. Kutueva, Nadezhda I. Aleksandrushkina, Boris F. Vanyushin, Denitsa R. Teofanova, And Lyuben I. Zagorchev

Submitted to section: Molecular Plant Sciences

The authors present a review on functional genomics of host-parasite interaction for a number of most investigated parasitic plant species (witchweed, broomrapes, and dodders). 

The major problem with this work is that it is not actually a synthesis of the data obtained on the subject. Rather the manuscript contains very long compilations (or synopses) of published papers on genomics and epigenomics of interactions between parasitic plants and their hosts. The manuscript is mostly compilation in nature. It is long retelling of what has been previously written on the subject. 

The structure of the manuscript is relatively monotonous. First, the authors indicate an article, which they are going to present. Then they retell the content of the article in detail often without any attempt to make comparisons with other studies. Finally, the authors formulate their own short commentary concerning the article presented. 

For instance: 

lines 115 – 165 (almost a full page of the manuscript) devoted to the analysis of the article #9 (Yoshida et al. 2019) with only a single shortly mentioned reference #12; 

lines 213 – 264 (almost a full page of the manuscript) devoted to the analysis of the article #15 (Cui et al. 2020) with only a single shortly mentioned reference #9;

lines 267 – 322 (almost a full page of the manuscript) devoted to the analysis of the article #16 (Kurotani et al. 2020) with no any additional references;

lines 325 – 368 (almost a full page of the manuscript) devoted to the analysis of the article #18 (Ogawa et al. 2021) with only a single shortly mentioned reference #11;

lines 380 – 406 (a half page of the manuscript) devoted to the analysis of the article #20 (Jiang et al. 2013) with no any additional references;

lines 407 – 473 (more than a full page of the manuscript) devoted to the analysis of the article #21 (Ranjan et al. 2014) with two other shortly mentioned references #22 and #23;

The same structure is continued up to the end of the manuscript excluding the short section “Conclusions” (one and half pages), which is look like real review. 

As a result of this approach, the manuscript is very long (50 pages) and resembles the "Literary Review" section of a dissertation, which details the preceding data on the subject under study. Unfortunately, the manuscript under consideration lacks the "Discussion" section, which would compare the results of different authors and form a general synthesis in the field of research on genomics and transcriptomics of interactions between parasitic plants and their hosts.

As a positive aspect I should mention, that the manuscript is clear and well written, with no fundamental flaws and weaknesses, and contains new and interesting data that are sound, adequately described, and that may provide important cues to scientists interested in molecular biology of parasitic plants. However, the style of the manuscript adopted by the authors raises some doubts in my mind as to its suitability for publication in a highly rated journal. Consequently, I leave it to the journal editor to decide whether to accept or reject this manuscript. 

Minor points: 

Lines 93-97: The authors state that higher ratios of non-synonymous to synonymous substitutions detected for the expanded gene families prove relaxed selection pressure, which is not correct. The relaxed selection should increase both, non-synonymous and synonymous substitutions proportionally. Rather, the higher ratios of non-synonymous to synonymous substitutions may indicate positive selection involved in evolution of the expanded gene families. 

Line 176: “de novo” should be in Italics. 

Line 192: The abbreviation “TF” should be deciphered.

Line 198:  β-expansin” should be in Italics.

Line 681: The abbreviation “hpi” should be deciphered.

Author Response

First of all, we want to thank the Reviewer for the thorough evaluation of our manuscript and for the valuable suggestions.

Following are our replies to the Reviewer’s specific comments and suggestions:

  1. The major problem with this work is that it is not actually a synthesis of the data obtained on the subject. Rather the manuscript contains very long compilations (or synopses) of published papers on genomics and epigenomics of interactions between parasitic plants and their hosts. The manuscript is mostly compilation in nature. It is long retelling of what has been previously written on the subject.

Reply: a couple of months ago we have been asked by the Editors of IJMS to contribute a comprehensive review. It looks like we have overdone with the comprehensiveness. Therefore, we have removed a significant part of the text mostly containing technical details that could be interesting to the people specifically dealing with genomics but are somewhat excessive for a general reader. The main text has become shorter and easier to read.

  1. The structure of the manuscript is relatively monotonous. First, the authors indicate an article, which they are going to present. Then they retell the content of the article in detail often without any attempt to make comparisons with other studies. Finally, the authors formulate their own short commentary concerning the article presented. 

For instance: 

lines 115 – 165 (almost a full page of the manuscript) devoted to the analysis of the article #9 (Yoshida et al. 2019) with only a single shortly mentioned reference #12; 

lines 213 – 264 (almost a full page of the manuscript) devoted to the analysis of the article #15 (Cui et al. 2020) with only a single shortly mentioned reference #9;

lines 267 – 322 (almost a full page of the manuscript) devoted to the analysis of the article #16 (Kurotani et al. 2020) with no any additional references;

lines 325 – 368 (almost a full page of the manuscript) devoted to the analysis of the article #18 (Ogawa et al. 2021) with only a single shortly mentioned reference #11;

lines 380 – 406 (a half page of the manuscript) devoted to the analysis of the article #20 (Jiang et al. 2013) with no any additional references;

lines 407 – 473 (more than a full page of the manuscript) devoted to the analysis of the article #21 (Ranjan et al. 2014) with two other shortly mentioned references #22 and #23;

The same structure is continued up to the end of the manuscript excluding the short section “Conclusions” (one and half pages), which is look like real review. 

As a result of this approach, the manuscript is very long (50 pages) and resembles the "Literary Review" section of a dissertation, which details the preceding data on the subject under study. Unfortunately, the manuscript under consideration lacks the "Discussion" section, which would compare the results of different authors and form a general synthesis in the field of research on genomics and transcriptomics of interactions between parasitic plants and their hosts.

Reply: We have deleted significant parts of the text noted above. Most of these deleted parts were describing technical info that could be interesting to people specifically involved in genomic studies but are probably excessive for a general reader. We have also deleted descriptions of some data that could be biologically relevant but whose functional significance is still obscure. Some technical details were transferred from the maim text to the Tables 1 and 2. Hopefully, in the present form the manuscript is shorter and more enjoyable to read. As concerning the “Discussion” section, we have made a long “Conclusions” section that fulfills the same tasks.

  1. As a positive aspect I should mention, that the manuscript is clear and well written, with no fundamental flaws and weaknesses, and contains new and interesting data that are sound, adequately described, and that may provide important cues to scientists interested in molecular biology of parasitic plants. However, the style of the manuscript adopted by the authors raises some doubts in my mind as to its suitability for publication in a highly rated journal. Consequently, I leave it to the journal editor to decide whether to accept or reject this manuscript. 

Reply: Thank You!

Minor points: 

  1. Lines 93-97: The authors state that higher ratios of non-synonymous to synonymous substitutions detected for the expanded gene families prove relaxed selection pressure, which is not correct. The relaxed selection should increase both, non-synonymous and synonymous substitutions proportionally. Rather, the higher ratios of non-synonymous to synonymous substitutions may indicate positive selection involved in evolution of the expanded gene families. 

Reply: We have stated that higher ratios of non-synonymous to synonymous substitutions were detected in the expanded gene families compared with the contracted gene families (where the non-synonymous substitutions are probably counter-selected). We think that the positive selection for non-synonymous substitutions could act at later stages of these genes’ evolution when their new functions were already acquired. To express this view more explicitly, we have added a statement to the respective text. 

  1. Line 176: “de novo” should be in Italics. 

Reply: This text has been deleted.

  1. Line 192: The abbreviation “TF” should be deciphered.

Reply: Done.

  1. Line 198:  “β-expansin” should be in Italics.

Reply: This sentence has been reformatted so that “β-expansin” became the name of a protein, not a gene. Therefore, no Italics font is needed.

Line 681: The abbreviation “hpi” should be deciphered.

Reply: Done.

Reviewer 3 Report

The review titled : “Genomic and epigenomic mechanisms of the interaction between parasitic and host plants”, submitted by the authors Ashapkin et al., reviewed the current understanding of how the host preferences are determined in different parasitic weeds, as well as the roles of both genetic and epigenetic features.

The review contains good amount of data about the selected topic and is if special interest for researchers within this field.

There are some things need to be addressed before the publishing of this review:

1.       The abstract and introduction are well written.

2.       The parasitic plant part is very long and need to be shortened, because its in the current status it looks like “a chapter in a book” and not an abstracted review article. The reader of a review need clear direct data well present as in tables or figures. So, its recommended here to shorten the description and convert it into tables summarizing the work on each parasitic species, hosted crops, method of parasitism, molecular background ..reference etc..

3.       If we go to the “Striga” in line 71, the data of the references from 9-12 need to reflected in an independent table, with more details than mentioned in the “1” table at the end of the review. In addition the data and description mentioned in lines 71-165 need to be shortened to 1-2 concise paragraphs.

4.       In the  Orobanche, in line 166 , its clear that there is a lot of work about Orobanchaceae family which you discuss here several species belonging to this family including Triphysaria versicolor and others, e.g. Phtheirospermum japonicum . This part shortening should be in single paragraph “small “about each species, mentioned molecular parasitism background and abstracting the recent research in this field in the form of a nice table including detailed data found for each.

5.       The same comment above applied to lines 378-787.

6.       The same comment in 4 applied to the title : Rafflesiaceae,  lines 982-1057

7.       In  title :Genetic basis of susceptibility and resistance in the host-parasite plant interaction, line 1093-1270 , Cowpea-Striga gesnerioide, this part should be shortened and data in it presented in table.

8.       The remaining of chapter is the same comment, you need to rearrange the data and presented in a much acceptable way.

I give you major revision.

Author Response

First of all, we want to thank the Reviewer for the thorough evaluation of our manuscript and for the valuable suggestions.

Following are our replies to the Reviewer’s specific comments and suggestions:

  1. The abstract and introduction are well written.

Reply: Thank You.

  1. The parasitic plant part is very long and need to be shortened, because its in the current status it looks like “a chapter in a book” and not an abstracted review article. The reader of a review need clear direct data well present as in tables or figures. So, its recommended here to shorten the description and convert it into tables summarizing the work on each parasitic species, hosted crops, method of parasitism, molecular background reference etc.

Reply: We have deleted significant parts of the text mostly containing the technical details and other information that could be excessive for a general reader not specifically interested in genomic studies. Most essential parts of this information were transferred to Tables 1 and 2. But we could not convert into tables all the info suggested by the Reviewer because of the large differences in the experimental setups between different studies. Anyhow, we think that in the present form this part of the manuscript became easier and more enjoyable to read. 

  1. If we go to the “Striga” in line 71, the data of the references from 9-12 need to reflected in an independent table, with more details than mentioned in the “1” table at the end of the review. In addition the data and description mentioned in lines 71-165 need to be shortened to 1-2 concise paragraphs.

Reply: Significant parts of the text in lines 71-165 were either transferred to the Tables 1 and 2 or deleted altogether.

  1. In the Orobanche, in line 166, its clear that there is a lot of work about Orobanchaceae family which you discuss here several species belonging to this family including Triphysaria versicolor and others, e.g. Phtheirospermum japonicum . This part shortening should be in single paragraph “small “about each species, mentioned molecular parasitism background and abstracting the recent research in this field in the form of a nice table including detailed data found for each.

Reply: Significant parts of this subsection were either transferred to the Tables 1 and 2 or deleted altogether.

  1. The same comment above applied to lines 378-787.

Reply: Significant parts of this subsection were either transferred to the Tables 1 and 2 or deleted altogether.

  1. The same comment in 4 applied to the title: Rafflesiaceae, lines 982-1057

Reply: Some parts of this subsection were either transferred to the Tables 1 and 2 or deleted altogether.

  1. In title: Genetic basis of susceptibility and resistance in the host-parasite plant interaction, line 1093-1270, Cowpea-Striga gesnerioide, this part should be shortened and data in it presented in table.

Reply: We have deleted significant parts of this subsection mostly containing the details that could be regarded as excessive by most readers.

  1. The remaining of chapter is the same comment, you need to rearrange the data and presented in a much acceptable way.

Reply: We have deleted significant parts of all remaining subsections mostly containing the details that could be regarded as excessive by most readers, but tried to keep all the information of high scientific interest and functional significance.

Round 2

Reviewer 1 Report

In general, authors revised the paper correctly. However, there are a few formatting mistakes. Authors should check the whole document for such mistakes after removing track changes. 

Author Response

Comment: In general, authors revised the paper correctly. However, there are a few formatting mistakes. Authors should check the whole document for such mistakes after removing track changes.

Reply: Thank you very much for your help in improving our paper. We have done a thorough checking and corrected all the mistakes found. 

Reviewer 2 Report

The revised version of the manuscript was significantly improved and now it is appropriate for the IJMS.

Author Response

Comment: The revised version of the manuscript was significantly improved and now it is appropriate for the IJMS.

Reply: Thank you very much for your help in improving our paper.

Reviewer 3 Report

Accepted for me 

Author Response

No comments have been made.

We thank the Reviewer for the great help in improving our paper.